# IFT proteins spatially control the geometry of cleavage furrow ingression and lumen positioning

Nicolas Taulet[1], Benjamin Vitre[1], Christelle Anguille[1], Audrey Douanier[1], Murielle Rocancourt[2], Michael Taschner[3], Esben Lorentzen[3], Arnaud Echard [2] & Benedicte Delaval [1]

Cytokinesis mediates the physical separation of dividing cells and, in 3D epithelia, provides a spatial landmark for lumen formation. Here, we unravel an unexpected role in cytokinesis for proteins of the intraflagellar transport (IFT) machinery, initially characterized for their ciliary role and their link to polycystic kidney disease. Using 2D and 3D cultures of renal cells, we show that IFT proteins are required to correctly shape the central spindle, to control symmetric cleavage furrow ingression and to ensure central lumen positioning. Mechanistically, IFT88 directly interacts with the kinesin MKLP2 and is essential for the correct relocalization of the Aurora B/MKLP2 complex to the central spindle. IFT88 is thus required for proper centralspindlin distribution and central spindle microtubule organization. Overall, this work unravels a novel non-ciliary mechanism for IFT proteins at the central spindle, which could contribute to kidney cyst formation by affecting lumen positioning.

[1] CRBM, CNRS, Univ. Montpellier, Centrosome Cilia and Pathology Lab, 1919 Route de Mende, 34293 Montpellier, France. [2] Institut PASTEUR, CNRS UMR 3691 Membrane Traffic and Cell Division Lab Cell Biology and Infection Department, 25-28 rue du Dr Roux, 75015 Paris, France. [3] Department of Molecular Biology and Genetics Aarhus University, Gustav Wieds Vej 10c, DK-8000 Aarhus C, Denmark. Correspondence and requests for materials should be addressed to B.D. (email: benedicte.delaval@crbm.cnrs.fr)

Cytokinesis leads to the separation of dividing cells[1] and is required, in 3D epithelia, to define the site of lumen formation[2–6]. It is initiated in anaphase, when the mitotic spindle reorganizes into dense arrays of antiparallel microtubules (MTs), the central spindle. This process is regulated both in space and time by the kinase Aurora B which is part of the chromosome passenger complex[7–9]. In anaphase, Aurora B is translocated to the central spindle by the kinesin MKLP2[10–13] and promotes the clustering and accumulation of the MKLP1/MgcRacGAP complex (centralspindlin)[13,14], by phosphorylating MKLP1[15]. This ultimately contributes to the local activation of the small GTPase RhoA, a key regulator of cleavage furrow ingression[7–9]. Altogether, these cytokinetic regulators serve to stabilize MT bundles, coordinate cleavage furrow ingression and ensure proper completion of cytokinesis.

Proteins of the intraflagellar transport machinery (IFT) are well-established and evolutionarily conserved regulators of cilia formation and function. In non-dividing ciliated cells, they function as part of transport complexes required for cilia formation and function[16–20]. Their disruption in animal models leads to kidney tubule defects similar to those seen in polycystic kidney disease[21–23]. Indeed, in mutant mice for *IFT88/Tg737*, a core member of the IFT machinery, an initially mild multifocal microscopic dilation of the proximal tubule is observed, rapidly followed by marked dilation of the lumen and by cyst formation[21]. However, the cellular mechanisms contributing to the initial defects in kidney tubule formation observed in animal models remain to be fully understood.

Even if cilia dysfunctions have been associated with kidney cyst formation, recent works indicate that IFT proteins could play, in addition to their ciliary functions, non-ciliary roles that may contribute to kidney cyst formation[24–29]. IFT proteins have been described to function in non-ciliary polarized transport[27,28] or in the regulation of MTs dynamics in the cytoplasm of interphase cells[26]. We also previously showed that IFT88 is required for proper spindle orientation in early mitosis[25]. Interestingly, IFT proteins were shown to concentrate at the cleavage furrow in *Chlamydomonas*, suggesting that they could also function during cytokinesis[30]. However, the role of IFT proteins during cytokinesis remains to be fully characterized. Furthermore, the contribution of IFT-related cytokinesis defects to kidney cyst formation has never been addressed.

We show that IFT88 plays an unexpected role during anaphase where it is required for Aurora B/MKLP2 relocalization to the central spindle, for the spatial control of cleavage furrow ingression and subsequently for proper kidney lumen positioning. Given that abnormal lumen formation is a key feature of kidney defects observed in IFT mutant mice[21], this work suggests a contribution of central spindle defects to abnormal lumen establishment in kidney diseases.

## Results

### IFTs are required for central spindle MT organization.
The established roles of IFT proteins in association with MTs in the cilium prompted us to test if they could function during anaphase in association with central spindle MTs. In anaphase, the core member of the IFT machinery, IFT88, diffusely localized along central spindle MTs (Supplementary Fig. 1a). To precisely localize endogenous IFT88 associated with MTs, we performed a proximity ligation assay (PLA) between IFT88 and α-tubulin in dividing LLC-PK1 kidney cells (Fig. 1a and Supplementary Fig. 1b). In anaphase, most of the PLA signal corresponding to IFT88 in association with α-tubulin is concentrated along central spindle MTs. This result suggests that IFT proteins could be required for MTs organization in anaphase.

To directly test this hypothesis, we used a siRNA-based approach to deplete IFT proteins from LLC-PK1 kidney cells. Proteins depletion was controlled by western blot and immunofluorescence (Fig. 1b) and central spindle organization was monitored with α-tubulin staining (Fig. 1c). Depletion of IFT88 or IFT27, two members of the IFT-B complex, severely disrupted the organization of central spindle MTs, compared to non-treated or siRNA control cells (Fig. 1c, d). This disorganization was confirmed using live imaging in LLC-PK1 cells expressing GFP-α-tubulin (Fig. 1e, Supplementary Movie 1). More specifically, IFT88-depleted cells showed wider and disorganized arrays of central spindle MTs, indicating defects in MTs bundling in anaphase. Importantly, central spindle MTs disorganization was rescued by expressing a mCherry-IFT27 construct not targeted by IFT27 siRNA, demonstrating the specificity of the phenotype (Fig. 1f). Similar defects were observed using independent human siRNA oligonucleotides in HeLa cells, further validating the specificity of the phenotype in a different, non-ciliated cell type (Supplementary Fig. 1c, d). These results indicate that IFT proteins are required to properly organize central spindle MTs.

Components of the centralspindlin complex, including the kinesin MKLP1 and MgcRacGAP, were previously described for their roles in MTs bundling and MTs-membrane interaction during anaphase[1,13,14]. We therefore tested if IFT proteins could participate in central spindle organization by contributing to the accumulation of centralspindlin components at the central spindle. The localization of MKLP1 and MgcRacGAP were monitored by immunofluorescence upon control or IFT proteins depletion. Depletion of either IFT27 or IFT88 impaired the localization of MKLP1 and MgcRacGAP in anaphase cells. Indeed, both proteins were decreased and not uniformly distributed across the central spindle (Fig. 1g, h, Supplementary Fig. 1e). Importantly, IFT depletion perturbed their localization without affecting their expression levels (Fig. 1i). In contrast, PRC1, another key regulator of cytokinesis[1], accumulated at the central spindle in both control and IFT-depleted cells (Supplementary Fig. 1f). Even if slightly mislocalized, most likely due to abnormal MTs organization in IFT-depleted cells, PRC1 was still present at the central spindle in anaphase (Supplementary Fig. 1f). Altogether, these results indicate that IFT proteins are required for the localization in anaphase of central spindle components, namely the centralspindlin complex MKLP1 and MgcRacGAP, but do not affect the localization of all cytokinetic components. Given the importance of these proteins in MTs bundling and cleavage furrow formation, IFT proteins likely participate in central spindle organization by contributing to their localization in anaphase.

### IFT88 is required for proper Aurora B relocalization.
To further investigate how IFT proteins influence central spindle MTs organization, we performed an unbiased mass-spectrometry approach on kidney cells enriched in mitosis to identify novel IFT mitotic interacting partners. IMCD cells stably expressing FLAG-tagged IFT27 were synchronized in mitosis and lysates were used for immunoprecipitation with a FLAG antibody followed by a FLAG peptide elution. Bioinformatics analysis of the mass-spectrometry results led to the identification of most of the members of the IFT-B complex as IFT27 interactors (Supplementary Table 1). This suggests that at least part of the IFT-B complex described in cilia is conserved in dividing cells. In agreement with this result, FLAG immunoprecipitations performed on mitotic shake-off of IMCD cells stably expressing FLAG-tagged IFT27, IFT52 or IFT46 confirmed these interactions in mitosis (Supplementary Fig. 2). Of note, all the IFT proteins identified belonged to the IFT-B complex but no

interaction was detected with the IFT-A member IFT140 (Supplementary Fig. 2). The proteomic analysis also led to the identification of Aurora B, a major cytokinetic regulator[7,9], as IFT-interacting partner (Supplementary Data 1). We thus hypothesized that IFT proteins could be required for the localization of the kinase to the central spindle and, subsequently, for the proper localization of MKLP1 and MgcRacGAP at this site. In agreement with proteomic results, Aurora B was found to form a complex with IFT proteins in mitotic cells (Fig. 2a, b). PLA experiments further revealed that IFT88 was associated with Aurora B along central spindle MTs during anaphase (Fig. 2c, upper panel). This staining was specific since IFT88 depletion abolished the signal (Fig. 2c, lower panel). Importantly, depletion of IFT27 or IFT88 impaired the Aurora B localization at the central spindle in

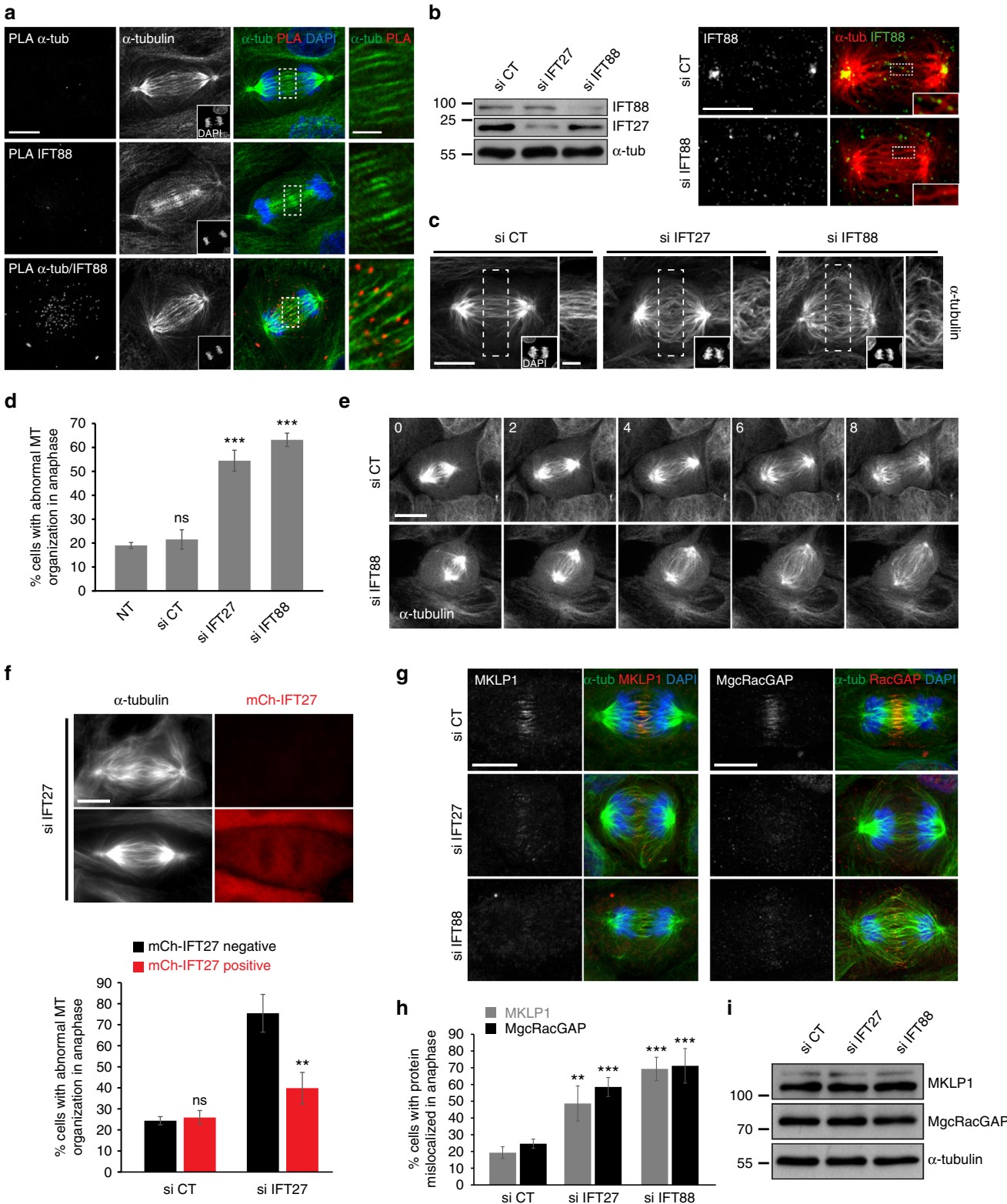

anaphase (Fig. 2d, e). Instead of being concentrated in the central spindle region, Aurora B staining diffusely spread between the chromosomes and the central spindle in around 50% of IFT-depleted cells, as compared to 15% in control cells. This diffuse staining indicates a defect in the relocalization of the kinase from the chromosomes to the central spindle during anaphase upon IFT proteins depletion (Fig. 2d, e). To further characterize this defect, we monitored Aurora B during drug-synchronized monopolar cytokinesis (Fig. 2f). This approach offers a powerful way to monitor the relocalization of cytokinetic regulators towards the cleavage furrow which corresponds, in this system, to the most polarized region of the cell[31,32]. This technique also ensures a similar timing after release and allowed to rule out the contribution of a defect in antiparallel MTs bundling in the abnormal relocalization of cytokinetic components. LLC-PK1 cells were synchronized in monopolar mitosis using a kinesin-5 inhibitor (STLC), then forced into monopolar cytokinesis by adding the Cdk1 inhibitor RO3306 (Fig. 2f). Under these conditions, IFT88 depletion was controlled by western blots (Fig. 2g). As observed on bipolar spindles, IFT88 depletion impaired Aurora B relocalization in monopolar cytokinesis (Fig. 2h, i). Indeed, while Aurora B was completely relocalized to the polarized region of the cell within 10 min after induction of monopolar cytokinesis in most polarized control cells, it was still diffusely distributed in approximately 50% of IFT88-depleted cells, further confirming a defect in the relocalization of the kinase (Fig. 2h, i). To ensure an accurate depletion of IFT88 in mitosis, we then combined the monopolar cytokinesis approach with an auxin inducible degron (AID) approach[33]. The AID tag was fused to the endogenous alleles of IFT88 using a CRISPR-Cas9 and IFT88 degradation was specifically induced in AID-IFT88 using auxin treatment in cells synchronized in monopolar mitosis. Degradation of AID-IFT88 was already observed after 1 h of treatment (Fig. 2j, upper panel). Under these conditions, IFT88 degradation in mitosis impaired Aurora B relocalization during monopolar cytokinesis (Fig. 2j, lower panel, Supplementary Fig. 3a). Given that IFT88 degradation is induced when cells are already synchronized, this degron approach allows us to rule out a potential contribution of earlier mitotic defects to the observed anaphase phenotype. Altogether, these results show that IFT88 is required for a timely relocalization of Aurora B towards the site of the cleavage furrow in early anaphase, at the time of central spindle establishment. This defect in Aurora B relocalization likely explains the decrease in MKLP1, phospho S708 MKLP1 (Supplementary Fig. 3b)[15] and MgcRacGAP observed at the central spindle and thus the central spindle MTs disorganization observed upon IFT depletion.

**IFT88 is required for Aurora B/MKLP2 relocalization.** Aurora B translocation from the kinetochores to the central spindle in anaphase depends on the mitotic kinesin-like protein 2 (MKLP2)[11,12]. To test whether IFT proteins could interact with MKLP2, we performed co-immunoprecipitation experiments between MKLP2 and IFT88 using a HeLa BAC line expressing GFP-MKLP2[34] under the control of its endogenous promotor. Using this approach on cells synchronized in mitosis, we found an interaction between MKLP2, Aurora B, and IFT88 (Supplementary Fig. 4a). To further confirm this interaction and assess whether IFT proteins could directly interact with MKLP2, we used two-hybrid assays. An interaction was detected between IFT88 and the C-terminal part (aa 530-887) of MKLP2 (Fig. 3a). No interaction was observed with the N-terminal part (motor domain, aa 1-529) of MKLP2 or with the Lex vector alone used as a negative control, thus demonstrating the specificity of the interaction (Fig. 3a). Using this approach, we also mapped the region of the kinesin interacting with IFT88 between amino acid 530 and 665 (IFT88-binding domain, BD; Fig. 3a, common between M1-Q665 and C-term constructs). Importantly, the interaction between the binding domain of MKLP2 and a sub-complex of IFT proteins (IFT88-IFT70-IFT52-IFT46)[18] was also confirmed in vitro using recombinant proteins (Fig. 3b and Supplementary Fig. 4b). These results indicate a direct interaction between IFTs and the non-motor part of MKLP2, raising the possibility that IFT88/MKLP2 might be important for Aurora B/MKLP2 relocalization to the central spindle.

To assess the effect of IFT88 depletion on the relocalization of Aurora B associated with MKLP2, we next took advantage of the monopolar cytokinesis approach and combined it to quantitative analysis of PLA. In control cells, PLA signal of the Aurora B/MKLP2 complex concentrated on central spindle MTs in bipolar division (Supplementary Fig. 4c) and was enriched in the most polarized region of the cell upon monopolar cytokinesis (Fig. 3c). In contrast, IFT88 depletion significantly reduced the percentage of Aurora B/MKLP2 PLA signal present in this region (Fig. 3c, d). This result shows that the relocalization of the Aurora B/MKLP2 complex is affected upon IFT88 depletion, since most of the PLA signal was still present in the "non-polarized region" (Fig. 3c, d). Of note, quantitative analysis of PLA signals showed that the overall number of PLA dots remained unchanged between control and IFT88-depleted conditions (Fig. 3e), suggesting that IFT88 depletion did not affect Aurora B/MKLP2 association.

To monitor the relocalization of MKLP2 in anaphase upon IFT88 depletion, live imaging was performed on a HeLa BAC line expressing GFP-MKLP2. In most control cells, GFP-MKLP2 translocated from the chromosomal region to uniformly decorate

**Fig. 1** IFT proteins associate with central spindle MTs in anaphase and are required for their organization. **a** Proximity ligation assay (PLA) showing the association between α-tubulin and IFT88 in anaphase LLC-PK1 cells. α-tubulin and IFT88 antibodies were used alone as negative controls. Maximum projections of PLA staining (left), α-tubulin (α-tub-FITC) middle panel and DAPI (inset) are shown. Magnifications, right: single confocal section of the boxed area at the central spindle. **b** Western blots (left) showing the amount of IFT27 and IFT88 in LLC-PK1 cells transfected with control (CT), IFT27 or IFT88 siRNA. α-tubulin: loading control. Immunofluorescence images (right) showing a decrease of IFT88 staining in an anaphase cell upon IFT88 depletion. **c** Immunofluorescence images of anaphase GFP-α-tubulin LLC-PK1 showing abnormal central spindle MTs organization in IFT27 and IFT88-depleted cells. Insets: DAPI. Maximum projections are shown. Magnifications, right: single confocal section of the dashed boxes. **d** Percentage of anaphase cells with abnormal central spindle MTs organization. n > 50 cells, 3 experiments. Mean +/− s.d. ns: not significant compared to non-treated (NT) condition and ***P < 0.001 compared to non-treated and control (t test). **e** Images from time-lapse microscopy showing the mitotic progression of GFP-αtubulin LLC-PK1 cells transfected with CT or IFT88 siRNA. Time (min). **f** Depletion of endogenous IFT27 (pig siRNA) is rescued by the expression of mCherry-IFT27 (human cDNA) resistant to IFT27 pig siRNA. Immunofluorescence images (top) showing central spindle MTs organization in LLC-PK1 cells depleted in IFT27 and expressing or not mCherry-IFT27. Graph (bottom): percentage of anaphase cells with MTs defects. n > 30 cells, 3 experiments. Mean +/− s.d. ns: not significant and **P < 0.01 compared to mCh-IFT27 negative cells (t test). **g** Immunofluorescence images of anaphase LLC-PK1 cells showing defects in MKLP1 and MgcRacGAP localization. α-tubulin/MKLP1/DAPI (left panel) and α-tubulin/MgcRacGAP/DAPI (right panel) stainings are shown. **h** Percentage of cells with MKLP1 or MgcRacGAP localization defects in anaphase. n > 50 cells. 3 experiments. Mean +/− s.d. **P < 0.01 and ***P < 0.001 compared to control (t test). **i** Western blots showing equal amounts of MKLP1 and MgcRacGAP in LLC-PK1 cells transfected with the indicated siRNA. α-tubulin: loading control. Scale bars: 10 μm for main images, 3 μm for insets

a confined region of the central spindle in anaphase (Fig. 3f, Supplementary Movie 2, Supplementary Fig. 4d). However, this distribution was affected in 55% of IFT88-depleted cells, where MKLP2 was either diffusely or asymmetrically distributed at the central spindle (Fig. 3f, Supplementary Movie 2, Supplementary Fig. 4d). To directly prove that IFT88 depletion can affect MKLP2 movement on MTs in anaphase, we performed Fluorescence Recovery After Photobleaching (FRAP) experiments on GFP-MKLP2 expressing HeLa cells. GFP-MKLP2 was photobleached in anaphase cells between the spindle pole and the central spindle (Fig. 3g, green box) and fluorescence recovery was monitored along MTs (Fig. 3g) and quantified (Fig. 3h). These experiments showed that IFT88 depletion significantly impaired fluorescence recovery of GFP-MKLP2 in anaphase cells (Fig. 3g, h), demonstrating that IFT88 is required for the proper relocalization of the motor along MTs towards the central spindle region. Overall, these experiments show that IFT88 is required for a timely and uniform relocalization of the Aurora B/MKLP2 complex to the central spindle via a direct interaction with the MKLP2 kinesin. By contributing to Aurora B relocalization, IFT88 is thus essential for proper localization of MKLP1 and MgcRacGAP, which in turn are required for normal central spindle MTs organization.

**The geometry of cleavage furrow ingression depends on IFT88.** Since central spindle organization is crucial for proper cleavage

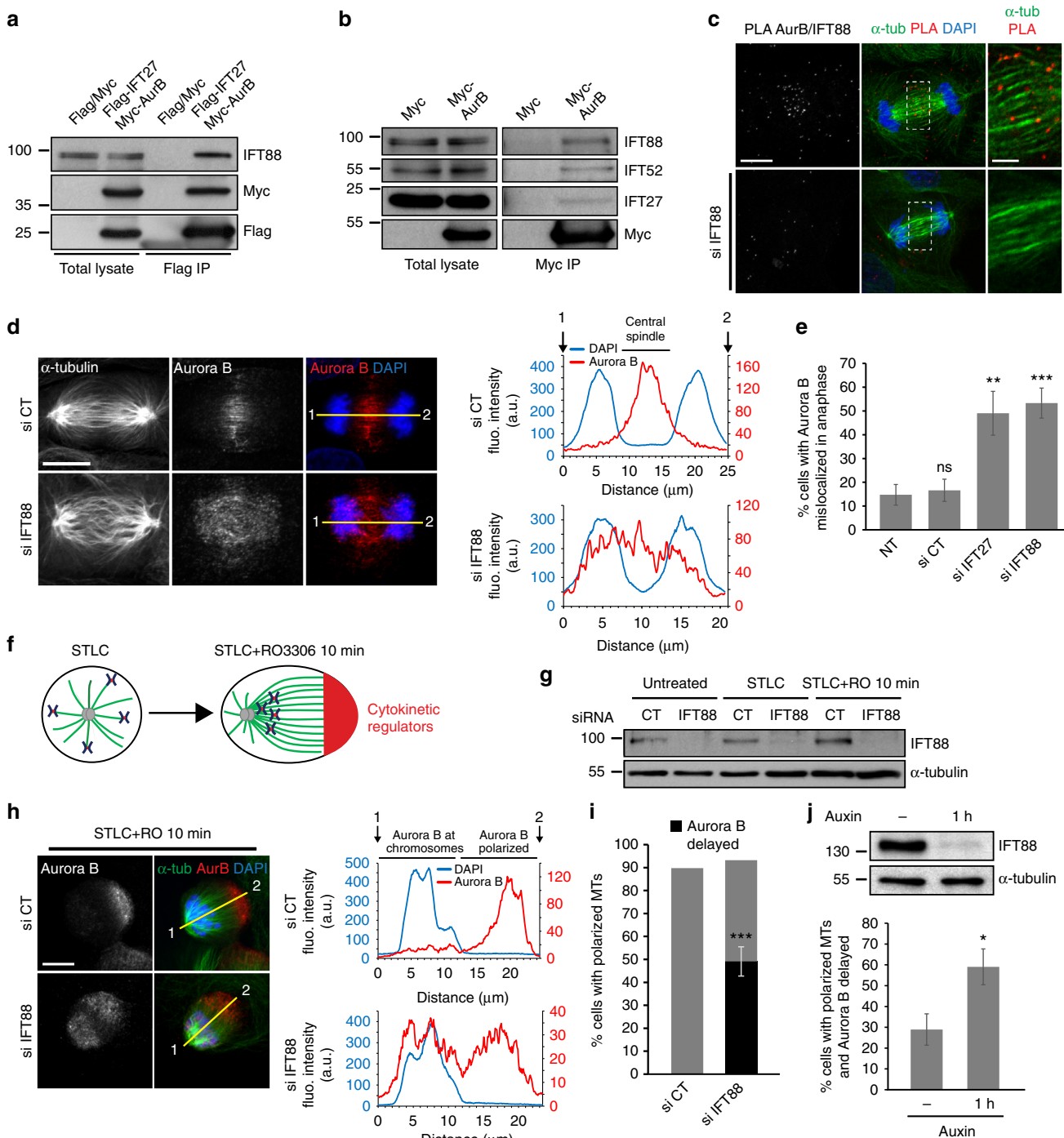

furrow ingression, we next monitored by live imaging how IFT88-depleted cells progressed through cytokinesis using a GFP-α tubulin/mCherry-H2B LLC-PK1 cell line (Fig. 4a, Supplementary Movie 3). In these cells, cleavage furrow ingression became highly asymmetric in a significant number of IFT88-depleted cells compared to control (Fig. 4a, b). Indeed, ingression occurred predominantly on one side of the central spindle in IFT88-depleted cells (Fig. 4c). Of note, an asymmetric localization of the small GTPase RhoA, a key regulator of cleavage furrow ingression, was observed[1] (Fig. 4d). Altogether, these results indicate that IFT88 is required to spatially control cleavage furrow ingression. Importantly, we observed that a low dose of paprotrain, a selective inhibitor of MKLP2 motor activity[35], both affected Aurora B relocalization to the central spindle (Supplementary Fig. 4e), and phenocopied the geometry defects in cleavage furrow ingression observed upon IFT88 depletion (Fig. 4e, f). This result strengthens the fact that IFT proteins can function with MKLP2 to ensure proper cleavage furrow ingression.

**IFT88 is required for lumen positioning in 3D renal cultures.** Defects in IFT proteins have been linked to polycystic kidney disease[17,21-23]. Indeed, in mice mutants for IFT88/Tg737, an initially mild multifocal microscopic dilation of kidney tubule was rapidly followed by a marked dilation of the lumen and by cyst formation[21]. To test if the cleavage furrow ingression defects observed in 2D culture upon IFT88 depletion could contribute to the initial defects in lumen formation observed in vivo, we took advantage of a 3D micropattern-based renal cell culture approach. When grown in 3D extracellular matrices, LLC-PK1 kidney cells form polarized spherical monolayers surrounding a central lumen[5,6,36,37]. Under these conditions, cytokinesis is essential to spatially control the establishment of a solitary central lumen by providing the spatial landmark for de novo formation of the lumen initiation site (Fig. 5a)[4,6,38].

To assess the effect of IFT88 depletion on lumen formation in 3D cultures, lumen initiation site positioning was monitored after completion of the first cell division using F-actin or the apical protein podocalyxin/GP135[5,6], a lumen-promoting factor marker. In siRNA control conditions, most two-cell stage 3D cultures established a central lumen, positioned between the two nuclei (Fig. 5b, upper left, c). However, 37% of IFT88-depleted cells showed defects in lumen initiation site positioning, as revealed by laterally localized F-actin (Fig. 5b, lower left, c) or podocalyxin (Fig. 5d) combined with mispositioned nuclei. Importantly, abnormal lumen positioning upon IFT88 depletion correlated with the presence of lateral cytokinetic bridge (Fig. 5e), showing

that defects in lumen positioning are associated with asymmetric cleavage furrow ingression. Importantly, a low dose of the MKLP2 motor inhibitor affecting Aurora B relocalization to the central spindle and inducing furrow ingression defects (Fig. 4e, f), phenocopied the lumen positioning defects observed upon IFT88 depletion (Fig. 5b, right panels, c). This result strengthened the fact that lumen defects could result from cytokinesis defects. Of note, even though MKLP2 staining was detected at the base of cilia, the MKLP2 inhibitor had no effect on cilia formation (Supplementary Fig. 5a-c). This indicates that the defects in lumen positioning observed upon MKLP2 inhibition do not result from a potential contribution of MKLP2 on cilia. At later stages of 3D cultures (three- to six-cell stages), lumen defects were also observed upon IFT88 depletion with a significant increase of multifocal lumens (Supplementary Fig. 6a,b), recapitulating what is observed in vivo when defects start to appear in kidney tubules. Altogether, these results show that IFT88 is required in 3D culture of epithelial renal cells to spatially control the geometry of cleavage furrow ingression and the subsequent establishment of a central lumen.

## Discussion
Collectively, our results show that IFT proteins, known mainly for their role in cilia, play an unexpected role during anaphase where they are required for the spatial control of symmetrical cleavage furrow ingression, which ultimately influences kidney lumen positioning.

Mechanistically, we show here that IFT88 is required for the proper relocalization of the Aurora B/MKLP2 complex to the central spindle, through a direct interaction with MKLP2. We propose that this contributes to central spindle organization by ensuring a timely and uniform distribution of the centralspindlin complex, then leading to localized activation of the small GTPase RhoA. IFT proteins thus contribute to the spatial control of cleavage furrow ingression (see model Fig. 6, left panels). By characterizing a novel molecular pathway involving IFT proteins in the relocalization of the Aurora B/MKLP2 complex to the central spindle in anaphase, this study also highlights that the IFT machinery contributes to several transport functions not limited to cilia. In anaphase, IFT proteins would still function in association with MTs, as described previously in ciliated[19,20] and non-ciliated systems[25-28]. However, we show here that IFT88, and most likely other IFTs of the same sub-complex, function in anaphase with a distinct MT associated motor, namely MKLP2. This work therefore sets the stage for future studies that will aim at characterizing in detail non-ciliary IFT/motor transport complexes. Of note, the two components of the centralspindlin

---

**Fig. 2** IFT proteins are required for the timely relocalization of the kinase Aurora B to the central spindle. **a** Flag immunoprecipitation performed on mitotic HeLa Kyoto cells transfected with Flag and Myc or with Flag-IFT27 and Myc-Aurora B. **b** Myc immunoprecipitation performed on mitotic HeLa Kyoto cells transfected with Myc or with Myc-Aurora B. **c** PLA between Aurora B and IFT88 on anaphase LLC-PK1. Cells depleted in IFT88, negative control. Controls for IFT88 or Aurora B alone are provided in Fig 1a and Supplementary Fig. 4c, respectively. Maximum projections are shown. Magnification, right: single confocal section of the dashed boxed area at the central spindle. **d** Immunofluorescence (left) showing α-tubulin or Aurora B staining in LLC-PK1 cells. Line scans (right) indicate whether Aurora B is accumulated at the central spindle or diffusely localized. Aurora B and DAPI fluorescence intensities are measured from 1 to 2 along the yellow line. **e** Percentage of anaphase cells with Aurora B mislocalization. $n > 50$ anaphase cells. 3 experiments. Mean $+/-$ s.d. ns: not significant compared to non-treated (NT) condition; $**P < 0.01$ and $***P < 0.001$ compared to non-treated and control ($t$ test). **f** Schematics: upon monopolar cytokinesis, cells polarize, reorganize their MTs (green) and cytokinetic regulators (red) are recruited to the most polarized area of the cell. STLC: kinesin-5 inhibitor S-trityl-l-cysteine. RO3306: Cdk1 inhibitor. **g** Western blots showing IFT88 depletion in LLC-PK1 cells treated or not with STLC then released or not in RO3306 for 10 min. α-tubulin: loading control. **h** Immunofluorescence images (left) showing Aurora B staining (red) in LLC-PK1 cells undergoing monopolar cytokinesis. α-tubulin (green) and DAPI (blue) are used to monitor cell polarization. Line scans (right) indicate whether Aurora B is well polarized or delayed. Fluorescence intensities are measured as in (**d**). **i** Percentage of polarized LLC-PK1 cells presenting a delay in Aurora B relocalization. $n > 150$ cells. Three experiments. Mean $+/-$ s.d. $***$, $P < 0.001$ compared to control ($t$ test). **j** Western blots (upper panel) showing AID-YFP-IFT88 depletion in HCT116 cells upon Auxin treatment (1 h). Percentage (lower panel) of polarized cells (STLC + RO3306 30 min) presenting a delay in Aurora B relocalization. $n > 100$ cells. Three experiments. Mean $+/-$ s.e.m. $*$, $P < 0.05$ compared to control ($-$) ($t$ test). Scale bars: 10 µm for main images, 3 µm for insets

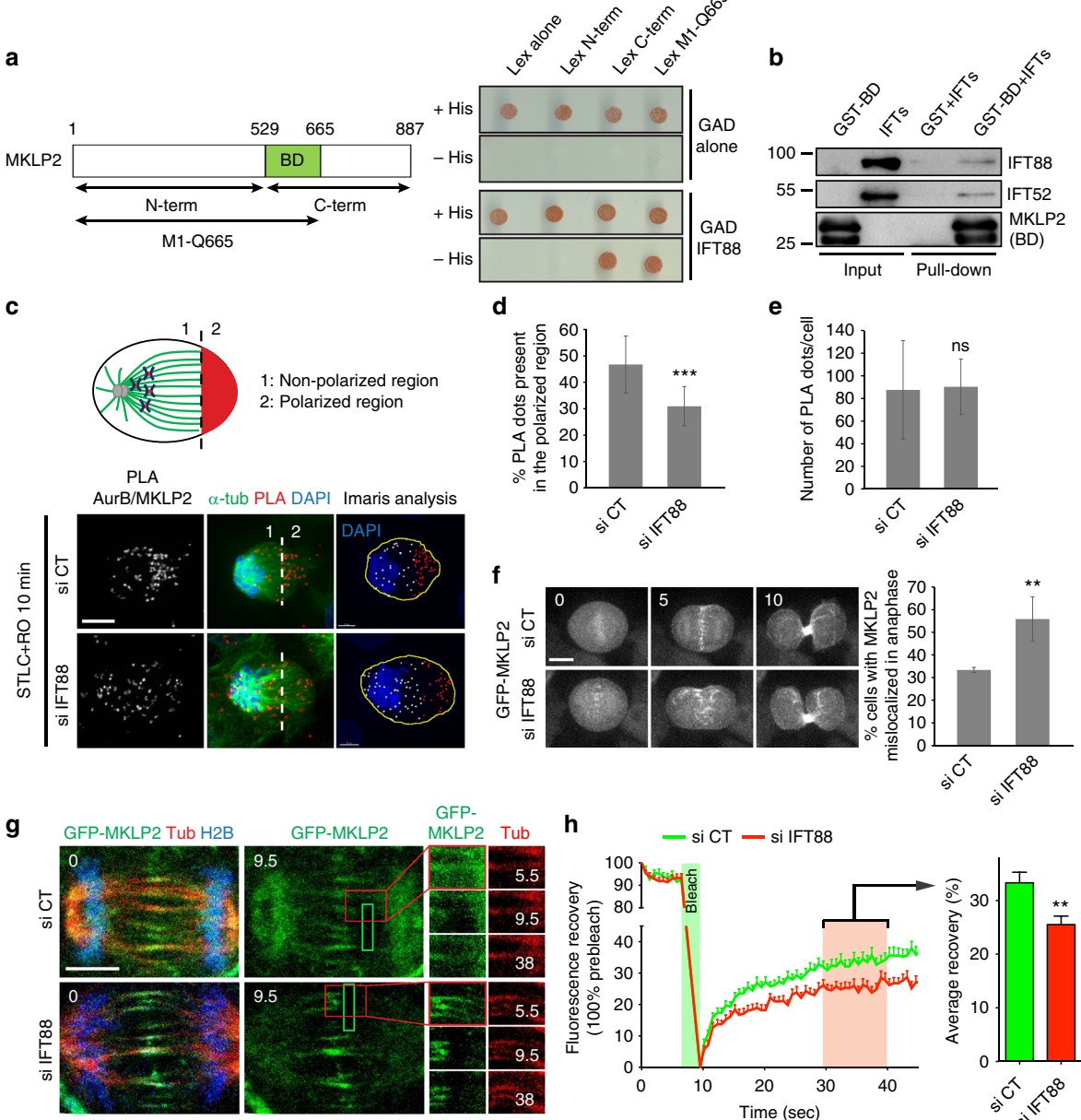

**Fig. 3** IFT88 interacts with MKLP2 and is required for a timely and uniform relocalization of the Aurora B/MKLP2 complex to the central spindle. **a** Specific interaction between IFT88 and MKLP2 in a yeast two-hybrid assay unraveling IFT88-binding domain (BD). Schematics of MKLP2 domains (left). *S. cerevisiae* reporter strain expressing the indicated GAD- and Lex- fusion proteins are grown on selective medium with or without Histidine (right). **b** Recombinant GST and GST-tagged MKLP2-BD bound to Glutathione sepharose beads were incubated with recombinant IFTs (IFT88-70-52-46). Interacting proteins were analyzed by western blots, unraveling an interaction between IFTs and MKLP2 (BD). **c** Schematic showing a monopolar cytokinetic cell divided into two regions: (1) non-polarized region and (2) polarized region where cytokinetic regulators accumulate. PLA between MKLP2 and Aurora B in monopolar cytokinetic LLC-PK1 cells showing the localization of the Aurora B/MKLP2 complex. Maximum projections are shown (left and middle panels). Detection of PLA dots in cells (yellow line) was done using Imaris (right panel). White objects, PLA dots in the non-polarized region (1). Red objects, PLA dots in the polarized region (2). DAPI staining (blue) is shown. **d**, **e** Percentage of PLA dots in the polarized region (**d**) and total number of PLA dots per cell (**e**). $n > 17$ cells per condition. Mean +/− s.d. ns: not significant and ***$P < 0.001$ compared to control ($t$ test). **f** Time-lapse of mitotic HeLa Kyoto cells expressing GFP-MKLP2 and transfected with control or IFT88 siRNA. Time (min). Percentage of cells with defects in GFP-MKLP2 localization during anaphase. $n > 40$ cells. Four experiments. Mean +/− s.d. **$P < 0.01$ compared to control ($t$ test). **g** FRAP analysis of Hela Kyoto GFP-MKLP2 cells in anaphase. Green boxes indicate the photobleaching zone, red boxes are magnified in the insets (right). Time of recovery is indicated in seconds. Inset 5.5 s, image before photobleaching. Inset 9.5 s, image after photobleaching. **h** Quantification of FRAP experiments. Green line, control. Red line, IFT88-depleted condition. Graph (right) shows the average recovery value measured between 30 and 40 s of recovery for individual cells. $n = 39$ cells for the control condition and $n = 47$ cells for the IFT88-depleted condition. Two experiments. Mean + s.e.m. **$P < 0.01$ compared to control ($t$ test). Time of recovery (seconds). Scale bars: 10 μm for main images, 3 μm for insets unless otherwise indicated

complex, MKLP1 and MgcRacGAP were also identified as IFT-interacting partners in the proteomic analysis. Further work will be required to assess whether IFT proteins could also directly regulate the localization of other motors or other central spindle proteins, including centralspindlin components.

Importantly, this work identifies for the first time a role for IFT proteins in controlling the positioning of cleavage furrow ingression during cytokinesis in kidney cells. IFT88 dysfunction was previously associated with abnormal cell ploidy in endothelial cells[39] and several IFT proteins were shown to concentrate at the cleavage furrow in *Chlamydomonas*[30], suggesting that they could function during cytokinesis. We show here that IFT88 is specifically required for proper positioning of cleavage furrow ingression in anaphase. It is however dispensable for completion of cytokinesis in this model. In agreement with this observation, MKLP1, MgcRacGAP and Aurora B were mislocalized in anaphase but concentrated at the midbody at later stages of cytokinesis in both control and IFT88-depleted cells (Supplementary Fig. 7). This indicates that IFT proteins are not required for their accumulation in late cytokinetic steps, but that they play a key role for their proper localization in anaphase to control the geometry of cleavage furrow ingression.

Finally, this work provides novel insights on how the geometry of cleavage furrow ingression can impact on lumen formation in 3D culture of kidney cells. We show here that IFT88 is essential to spatially control the establishment of a solitary central lumen in 3D culture systems. Given that cytokinesis provides the spatial landmark for the *de novo* lumen formation[3,6], we propose that the defects in lumen initiation site positioning observed at two-cells stage upon IFT88 depletion (Fig. 6, right panels) are caused by defects in the spatial control of cleavage furrow ingression. This result is strengthened by the fact that the use of MKLP2 inhibitor, which at low dose induces similar defects in the geometry of cleavage furrow ingression but does not interfere with cilia formation, phenocopies the lumen defects observed upon IFT88 depletion. Given that abnormal lumen formation is a key feature of kidney defects observed in IFT88 mutant mice[21], this work provides clues for the contribution of central spindle defects to abnormal lumen establishment in kidney diseases involving IFT proteins dysfunction. The multifocal lumen formation observed upon IFT88 depletion in three- to six-cell stage 3D cultures recapitulates what is observed at the initial stage of the disease in vivo. Our work thus suggests that cytokinesis defects could contribute to the initial multifocal microscopic dilation of proximal tubules observed in IFT88 mutant mice[21]. Further work will be required to assess whether the pathway outlined here for IFT88 also applies to other cilia proteins involved in human renal diseases[26].

## Methods

**Cell culture.** LLC-PK1 (CLS Cell Lines Services GmbH, Germany), GFP α-tubulin LLC-PK1 (Gift from P. Wadsworth[40]) were grown in a 1:1 mixture of Opti-MEM/HAM's F10 media supplemented with 10% fetal bovine serum (FBS). GFP-αtubulin/mCherry-H2B LLC-PK1 cells were generated by retroviral infection of GFP-αtubulin LLC-PK1 with a mCherry-H2B retrovirus and cells were sorted based on their fluorescence. HCT116 cells were grown in DMEM Glutamax medium supplemented with 10% FBS. HeLa Kyoto, GFP-MKLP2 HeLa Kyoto (Gift from A. Hyman to A. Echard)[34] and Flag-IFT IMCD cell lines (Gifts from G. Pazour[41]) were grown as described by American Type Culture Collection (Manassas, VA). mRFP-Histone H2B was introduced in the GFP-MKLP2 HeLa Kyoto cells using retroviral delivery and stable integrates were selected using 1 μg/ml puromycin.

**Cell synchronization and monopolar cytokinesis.** For immunofluorescence experiments in LLC-PK1 and HeLa Kyoto, cells were synchronized in G1/S 6 h post transfection using 2 mM thymidine (Calbiochem) for 16 h, then released from the thymidine block and fixed 9 h later to allow for anaphase enrichment. For immunoprecipitation experiments, HeLa Kyoto cells were synchronized in mitosis using nocodazole 100 ng/ml (Sigma-Aldrich) for 15 h followed by 1 h release or double thymidine (2 mM) block followed by 9 h release to reach 80% of mitotic

cells, including an average of 50% of anaphase. For the proteomic approach, IMCD cells were synchronized in G2/M phase using the Cdk1 inhibitor RO-3306 (10 μM, 15 h) (Calbiochem) then released for 1 h to ensure mitotic enrichment. For immunoprecipitation experiments, IMCD mitotic cells were isolated using mechanical shake-off. To induce monopolar cytokinesis, LLC-PK1 and HCT116 cells were treated with the kinesin-5 inhibitor S-trityl-l-cysteine (STLC, Santa Cruz) 2 μM for 15 h to generate monopolar cells then forced into cytokinesis by addition of Cdk1 inhibitor RO-3306 (2 μM) for 10 min (LLC-PK1) or 30 min (HCT116).

**siRNA and cDNA transfections.** Targeted proteins were depleted with small-interfering RNAs (siRNAs) designed and ordered via Dharmacon (Lafayette, CO) and delivered to cells at a final concentration of 100 nM using Oligofectamine (Invitrogen, Carlsbad, CA) according to manufacturers' instructions. Several siRNA sequences were used to target porcine IFT88: 5′-CCUUGGAGAUCGA-GAGAAUU-3′ and IFT27: 5′-GGGUGGAUCUGGUGGUGAAUU-3′ or human IFT88: 5′-CGACUAAGUGCCAGACUCAUU-3′ and IFT27: on-target plus SMARTpool. The efficacy of IFT proteins knockdown was assessed by immuno-blotting 48 h post transfection. Rescue experiment was performed by depleting endogenous porcine IFT27 in LLC-PK1 cell line expressing a human mCherry-IFT27 cDNA. cDNA transfections were performed using JetPEI (Polyplus Transfection) or Fugene 6 (Promega) transfection reagents according to manufacturers sinstructions for 24 h. cDNAs transfected include pCMV14-IFT27-3xFlag, pmCherry N1-IFT27, pCMV14-3xFlag and pcDNA3-Myc (provided by Montpellier Genomic Collection), pEGFP-C3 (Gift from N. Morin), GFP-MKLP2 and GFP-Podocalyxin (Gifts from A. Echard[5]) and Myc-Aurora B (Gift from D. Birnbaum[42]). The pBABE mCherry-H2B plasmid used for retrovirus production in HEK293T cells was a gift from K. Hached (T. Lorca/A. Castro lab).

**Lysates and immunoblotting.** HeLa Kyoto, LLC-PK1, IMCD and HCT116 cell extracts were obtained after lysis with buffer containing 50 mM Hepes (pH 7.5), 150 mM NaCl, 1.5 mM MgCl$_2$, 1 mM EGTA, 1% IGEPAL CA-630 and protease inhibitors (Sigma-Aldrich). Protein concentration for lysates was determined using Bradford reagent (Sigma-Aldrich), loads were adjusted, and proteins were resolved by SDS-PAGE and analyzed by western blot (Western Lightning Plus-ECL kit; PerkinElmer). Full blots are provided as Supplementary Figs. 8 and 9.

**Proteomics.** For each condition, the same amount of proteins (20 mg) was incubated with 40 μl Flag M2 agarose beads (Sigma-Aldrich) for 3 h at 4 °C. Beads were washed five times with 1 ml lysis buffer (50 mM Hepes (pH 7.5), 150 mM NaCl, 1.5 mM MgCl$_2$, 1 mM EGTA, 1% IGEPAL CA-630 and protease inhibitors). Flag-tagged proteins were then eluted with 0.5 mg/ml of Flag peptide (Sigma-Aldrich) for 30 min on ice. Samples were loaded on a SDS-PAGE and analyzed by mass-spectrometry on the Functional Proteomic Platform (FPP) using a LTQ-Orbitrap XL (Thermo Fisher Scientific). Proteins identification was performed with the Mascot software (Matrix Science).

**Immunoprecipitations.** For Flag and Myc immunoprecipitations, HeLa Kyoto cells were transfected with Flag and/or Myc empty vectors or with Flag-, Myc-tagged plasmids for 24 h and synchronized in mitosis. The resulting cell extracts were incubated for 2 h at 4 °C with Flag M2 agarose beads (Sigma-Aldrich) or with Myc-Trap A beads (Chromotek). For GFP immunoprecipitation, HeLa Kyoto cells transfected with GFP empty vector for 24 h or HeLa Kyoto stably expressing GFP-MKLP2 were synchronized in mitosis. Cell extracts were then incubated for 2 h at 4 °C with GFP-Trap A beads (Chromotek). In all cases, beads were washed four times with 800 μl lysis buffer and the immunoprecipitated proteins were separated by SDS-PAGE and analyzed by western blotting.

**Recombinant protein purification and GST pull-down assay.** The IFT88/70/52/46 complex was provided by M. Taschner and E. Lorentzen[18]. Briefly, the four expression plasmids were co-expressed in bacteria BL21 (DE3). Proteins were then affinity-purified by sequential passages over columns and tags were removed using TEV protease.

GST-BD (gift from A. Echard/S. Miserey-Lenkei) was expressed in the BL21 (DE3) Codon Plus strain of *Escherichia coli* after induction with 0.1 mM IPTG overnight at room temperature. Cells were lysed in buffer (20 mM K-PIPES pH 6.8, 1 mM MgCl$_2$, 1 mM EGTA, 100 mM NaCl, 1 mM DTT and protease inhibitors) containing 1 mg/ml lysozyme by sonication on ice. The GST fusion protein was affinity-purified using Glutathione Sepharose 4 fast flow beads (GE Healthcare) and eluted with 50 mM HEPES pH 7.5, 150 mM NaCl, 1 mM MgCl$_2$ and 10 mM reduced glutathione. GST was a gift from A. Bailly (D. Xirodimas lab).

For pull-down experiments, Glutathione Sepharose 4 fast flow beads (GE Healthcare) were blocked in pull-down buffer (80 mM PIPES, 2 mM MgCl$_2$, 0.5 mM EGTA, 250 mM NaCl, 1 mM DTT, 1% IGEPAL CA-630 and protease inhibitors) supplemented with 5% FBS for 1 h at 4 °C. Beads were washed two times and 5 μg of GST-BD or GST alone were loaded onto 10 μl of beads in pull-down buffer for 1 h at room temperature. Beads were washed three times and then incubated with 2.5 μg of a sub-complex of recombinant IFTs (IFT88-70-52-46) in pull-down buffer (200 μl) for 1 h at room temperature. Beads were washed five

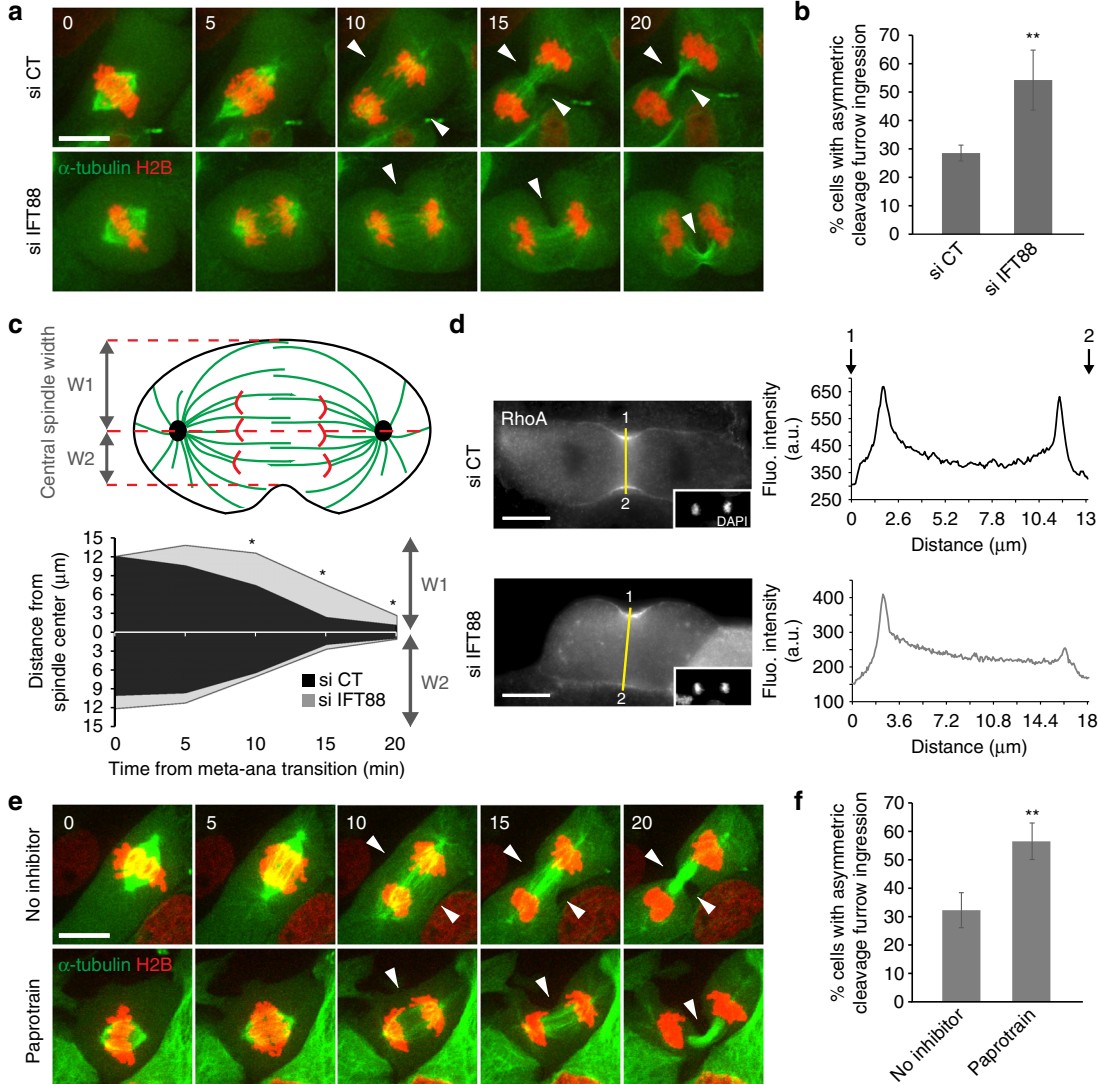

**Fig. 4** IFT88 is required for symmetrical cleavage furrow ingression in kidney cells. **a** Images from time-lapse microscopy showing cytokinesis progression of LLC-PK1 cells stably expressing GFP-α-tubulin and mCherry-H2B. White arrowheads indicate the position of cleavage furrow ingression. Time (min). **b** Percentage of cells undergoing asymmetric cleavage furrow ingression as shown in 4a. $n > 31$ cells. Three experiments. Mean +/− s.d. **$P < 0.01$ compared to control ($t$ test). **c** Schematics of an anaphase cell exhibiting asymmetric cleavage furrow ingression (upper panel) as observed in IFT88-depleted cells. Grey arrows define the central spindle width (W1, larger side where the ingression is delayed; W2 shorter side where the ingression occurs properly). Quantification of cleavage furrow ingression asymmetry over time (bottom graph). W1 and W2 distances are measured from the center of the central spindle during the metaphase to anaphase transition in control (black) or IFT88-depleted cells (grey). $n = 8$ cells. For W1 *$P < 0.05$ compared to control ($t$ test). **d** Immunofluorescence images showing RhoA localization in anaphase LLC-PK1 cells transfected with control or IFT88 siRNA. DAPI (inset). Line scans (right), RhoA fluorescence intensity along the yellow line measured from 1 to 2. **e** Images from time-lapse microscopy showing cytokinesis progression of LLC-PK1 cells (GFP-αtubulin/mCherry-H2B) treated or not with MKLP2 inhibitor (Paprotrain). White arrowheads indicate the position of cleavage furrow ingression. Time (min). **f** Quantification of the percentage of cells undergoing asymmetric cleavage furrow ingression. $n > 32$ cells. Three experiments. Mean +/− s.d. **$P < 0.01$ compared to control ($t$ test). Scale bars: 10 μm

times (200 μl) and resuspended in Laemmli buffer. Pulled-down proteins were detected by Coomassie blue staining and western blot.

**Yeast two-hybrid experiments**. Yeast two-hybrid experiments were performed by co-transforming the *Saccharomyces cerevisiae* reporter strain L40 with either pGAD-IFT88 or pGAD alone together with either pLex-mMKLP2 N-term (aa 1-529), pLex-mMKLP2 C-term (aa 530-887), pLex-mMKLP2 M1-Q665 (aa 1-665) or pLex alone. Transformed yeast colonies were selected on DOB agarose plates without Tryptophane and Leucine. Colonies were picked and grown for 3 days on DOB agar plates with Histidine to select co-transformants and without Histidine to detect interactions.

**Antibodies**. The following primary antibodies were used (western blot WB, immunofluorescence IF): IFT88 #13967-1-AP (WB: 1/500, IF: 1/250), IFT52

#17534-1-AP (WB: 1/500), IFT57 #11083-1-AP (WB: 1/250), IFT81 #11744-1-AP (WB: 1/250), IFT140 #17460-1-AP (WB: 1/500) from Proteintech, IFT27 (Novus Biologicals #NBP1-87170, WB: 1/200), MgcRacGAP (Abcam #ab2270, WB: 1/500, IF: 1/1000), α-tubulin (DM1α, Sigma-Aldrich #T6199, WB: 1/400), FITC-conjugated α-tubulin (DM1α, Sigma-Aldrich #F2168, IF: 1/300), Flag (Sigma-Aldrich #F7425, WB: 1/400), mCherry (GeneTex #GTX128508, IF: 1/300), RhoA (Santa Cruz 26C4 #sc-418, IF: 1/200), GFP (Roche #11814460001, WB: 1/500), Myc (9E10, Sigma-Aldrich #M4439, WB: 1/2000), MKLP1 (N-19, Santa Cruz #sc-867, IF: 1/500 and WB: Gift from D. Fesquet): 1/2000), Aurora B (BD Transduction laboratories #611082, WB: 1/500, IF: 1/250), MKLP2 (Gift from A. Echard, WB: 1/2000, PLA: 1/400), PRC1 (C-1 Santa Cruz #sc-376983, IF: 1/250), ARL-13 (Proteintech #17711-1-AP, IF: 1/250), Phospho S708 MKLP1 (Gift from M. Mishima, IF: 1/50)[15], Polyglutamylated tubulin (GT335, Gift from C. Janke/G. Bompard, IF: 1/200), Rhodamine Phalloidin (Cytoskeleton #PHDR1, IF: 1/300) and DAPI (Cell signaling, IF: 1/10000). Secondary antibodies include for IF: Alexa Fluor

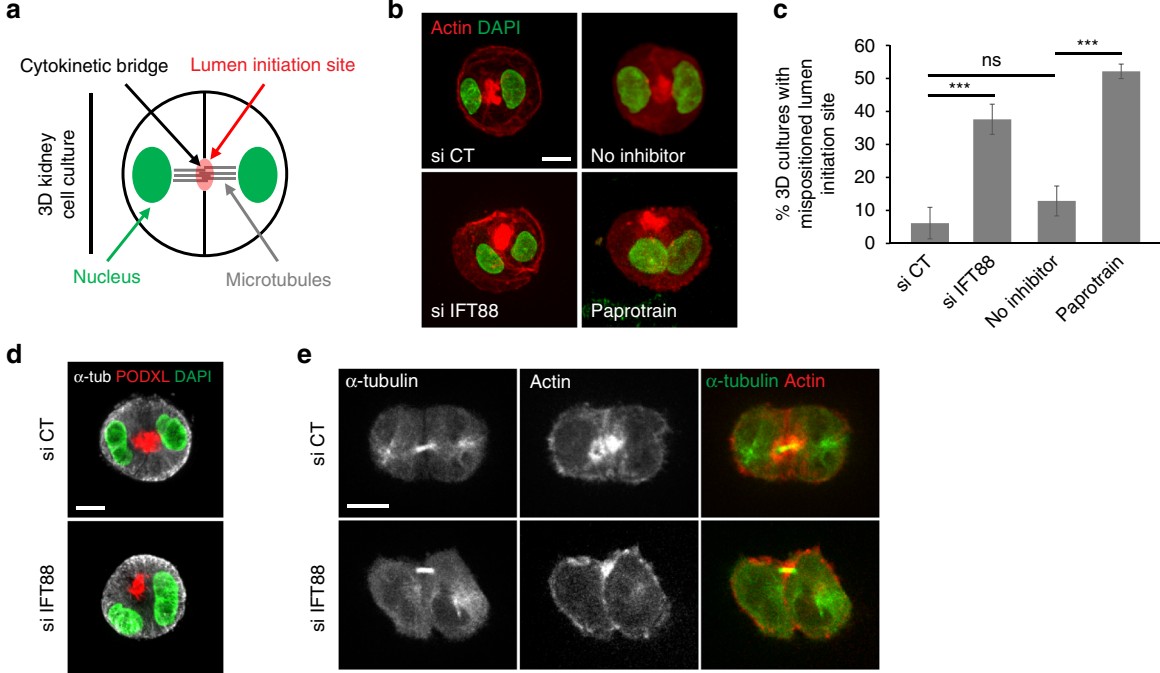

**Fig. 5** IFT88 is required for lumen positioning in 3D renal cultures. **a** Schematics of kidney cells organization in 3D culture at the transition from one to two-cell stage. The cytokinetic bridge, where antiparallel MTs (grey) overlap, defines the spatial landmark for the lumen initiation site (red). Under control condition, the lumen precursor (red) is centered between the two nuclei (green). **b** LLC-PK1 cells, depleted or not in IFT88 (left) and treated or not with Paprotrain (right), were grown in 3D on disk-shaped micropatterns for 24 h. Lumen positioning was monitored at two-cell stage, after the first cell division, using actin (red). DAPI is shown in green. **c** Percentage of two-cell stage 3D cultures with mispositioned lumen initiation site as shown by both lateral actin accumulation and nuclei mispositioning. $n > 30$ two-cells stage. 3 experiments. Mean +/− s.d. ns: not significant compared to control; ***$P < 0.001$ compared to control siRNA and no inhibitor ($t$ test). **d** Lumen positioning was detected in LLC-PK1 cells depleted or not in IFT88 and grown in 3D using the lumen-promoting factor podocalyxin (PODXL: red). α-tubulin (grey) and DAPI (green) are shown. **e** Images showing that lumen mispositioning (actin, red) correlates with lateral cytokinetic bridge (α-tubulin, green). Scale bars: 10 μm

488 (#4412S or #4408S), 555 (#4413S or #4409S) or 647 (#4414S or #4410S) -conjugated anti-rabbit or anti-mouse secondary antibodies (Molecular Probes, 1/1500) or Cy3-conjugated anti-goat (Jackson ImmunoResearch #705-165-147, 1/500) and for WB: anti-mouse and anti-rabbit IgG, HRP linked antibody (Cell signaling #7076 and #7074, 1/5000) and anti-goat IgG, HRP linked antibody (Jackson ImmunoResearch #805-035-180, 1/5000).

**Generation of HCT116 IFT88-AID cells**. HCT116 IFT88-AID cells were generated by adding an AID tag followed by a YFP tag at the 3′ end of the last exon on the IFT88 genomic locus. In detail, retroviral transduction was used to generate HCT116 cells stably expressing TIR1-9xMyc protein[33] and a clonal cell line was established. sgRNA targeting two regions adjacent to the 3′ end of IFT88 gene were introduced under the control of U6 transcription promoter into two separate vectors encoding for the expression of the Cas9 nickase (D10A)[43] (addgene 42335). A donor construct containing ≈ 600 bp recombination arms surrounding the 3′ end of IFT88 locus, in frame with a sequence encoding for an AID-YFP-Stop sequence, was generated. All three vectors were transfected into HCT116 TIR1 cells using Xtreme Gene 9 DNA transfection reagent (Roche). Cells were sorted based on their YFP fluorescence and single clones were isolated. Homozygous targeted clones were identified by PCR. Targeting of IFT88 and degradation of IFT88-AID-YFP was confirmed by western blot following addition of Auxin (Sigma-Aldrich) at 500 μM in the culture medium for 1 h.

**Immunofluorescence and PLA**. The cells were fixed in 4% paraformaldehyde in PHEM buffer (25 mM HEPES, 10 mM EGTA, 60 mM PIPES, 2 mM MgCl₂, pH = 6,9) for PLA and 3D experiments or −20 °C MeOH for immunofluorescence experiments to preserve MTs staining. For RhoA staining, a trichloroacetic acid (TCA) 10% for 10 min at 4 °C fixation protocol was used. Then, cells were blocked with PBS-BSA 1%-Triton 0.5% and stained for immunofluorescence with the appropriate primary and secondary antibodies. Slides were mounted in prolong gold (Life Technologies). PLA was performed according to the manufacturer's protocol (Duolink II Fluorescence, Duolink In Situ Detection Reagents Red) and PLA probes rabbit PLUS and mouse MINUS were used. Cells were fixed in 4% paraformaldehyde in PHEM buffer, then permeabilized and blocked with PBS-BSA 1%-Triton 0.5%. Primary mouse and rabbit antibodies were used alone as negative controls. Then, samples were incubated with the respective PLA probes for 1 h at

37 °C, ligated for 30 min at 37 °C and amplification with polymerase was performed for 100 min at 37 °C. To identify mitotic stages, cells were then stained for immunofluorescence with a FITC-conjugated α-tubulin antibody and DAPI before mounting in prolong gold. Images were acquired on a confocal microscope. PLA signal was quantified from 3D reconstructions with Imaris software (Bitplane).

**3D LLC-PK1 cultures**. LLC-PK1 cells expressing or not GFP-podocalyxin were transfected with siRNA control or IFT88 and used for 3D culture[36,37]. Twenty-four hours post transfection, cells were trypsinized and seeded (3000 cells per well) on disk-shaped micropatterns coated with collagen I 20 μg/ml (Sigma-Aldrich) in glass-bottom 96-wells microplate (CYTOO). 5 h after seeding, matrigel 2% (Corning) was added to the cells for 24 h to enrich the 3D cultures in two-cell stage or for 48 h to enrich in three- to six-cell stage. For MKLP2 inhibition, 5 μM of paprotrain (Calbiochem) was added when cells were seeded on micropatterns. Cells were fixed in 4% paraformaldehyde in PHEM buffer and processed for immunofluorescence. Images were acquired on a spinning disk confocal microscope.

**Cilia formation assay**. Primary cilia formation was induced in LLC-PK1 or RPE cells using medium containing 0.25% FBS for 48 h. Cells were fixed with −20 °C MeOH and stained with ARL-13 or Polyglutamylated tubulin to visualize cilia. Cilia formation was monitored in RPE cells treated or not with 5 μM paprotrain (Calbiochem) or 50 μM ciliobrevin D (Calbiochem) for 48 h in culture medium with 0.25% FBS.

**Microscopy and image analysis**. Epifluorescence images of the mCherry-IFT27 rescue experiment were acquired with a Leica DM6000 microscope (Objective: 63×/1.4 NA Plan-Apo) equipped with a Cool SNAP HQ2 camera and controlled by MetaMorph (Molecular Devices). Confocal images were acquired with a Zeiss LSM780 microscope (Objective: 63x/1.4 NA DIC Plan-Apo) controlled by ZEN software (Zeiss). Images of 3D cultures and time-lapse were performed using a spinning disk confocal microscope, a Nikon Ti Eclipse coupled to a Yokogawa spinning disk head and an EMCCD iXon Ultra camera (Objectives 60×/1.4 NA and 100×/1.45 NA), controlled by the Andor iQ3 software (Andor). Overnight time-lapse was started 30 h after siRNA transfection or 2 h after treatment with paprotrain 5 μM (Calbiochem) and images were acquired every 1, 2 or 5 min. Image processing and analysis (cropping, rotating, brightness, contrast adjustment,

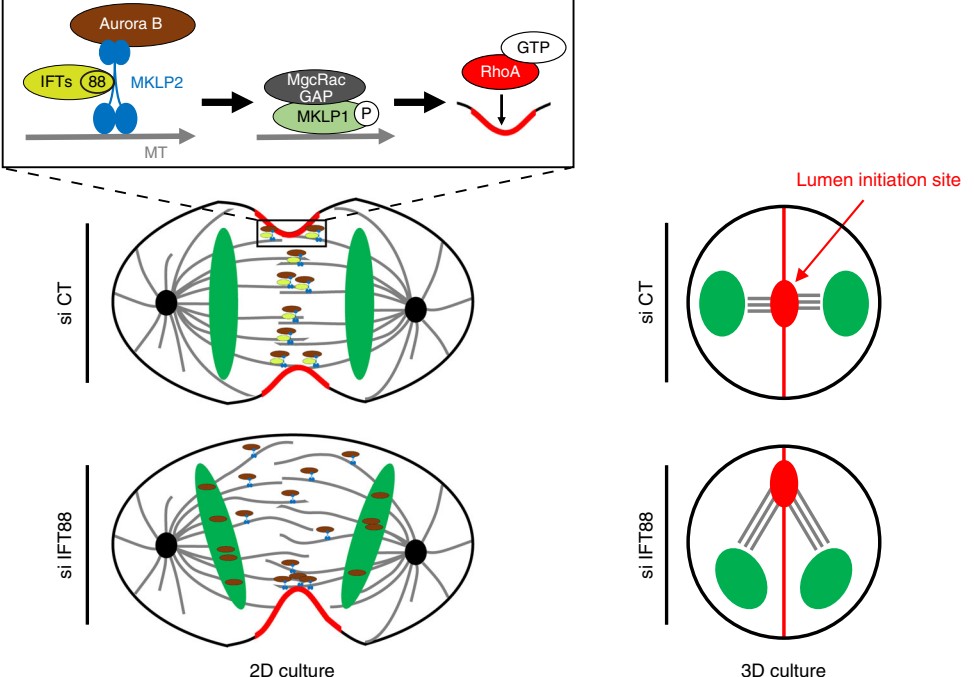

**Fig. 6** Molecular model for IFT function in central spindle organization, symmetrical cleavage furrow ingression and lumen positioning. IFT88 is depicted in this model as a component required for a timely and uniform relocalization of the Aurora B/MKLP2 complex to the central spindle through its interaction with MKLP2. It is thus essential for the relocalization of MKLP1 and MgcRacGAP, which are required for proper central spindle MTs assembly. By ensuring a timely and uniform distribution of cytokinetic regulators across the central spindle, including the localization of the small GTPase RhoA, IFT proteins spatially control symmetrical cleavage furrow ingression. Given that cytokinesis provides the spatial landmark for de novo-formed lumen, IFT88 would be required for proper lumen positioning in two-cell stage 3D cultures by controlling the geometry of cleavage furrow ingression

color combining and measurements) were performed with ImageJ or Imaris (Bitplane). Linescans were obtained using ImageJ plot profile tool. MKLP1, MgcRacGAP and Phospho S708 MKLP1 fluorescence intensities were measured using MetaMorph (Molecular Devices).

**FRAP experiments**. FRAP was performed on GFP-MKLP2 HeLa Kyoto cells expressing Histone H2B-RFP to facilitate the detection of early anaphase stages. Cells were seeded 24 h before the experiment in 35 mm glass-bottom culture dishes (Ibidi). 2 h before starting imaging, SiR-tubulin Cy5 (Spirochrome) at 100 nM was added to the medium to visualize MTs. Images were collected at 37 °C with a Zeiss LSM780 confocal microscope (Objective: 63×/1.4 NA DIC Plan-Apo). FRAP experiments were performed using the 488 nm laser line controlled by ZEN software (Zeiss). Images were acquired on a single plan every 500 ms, 9 acquisitions were done pre-bleach. The bleached zone was a fixed area represented by the green square in Fig. 3g. FRAP analysis was performed, using the ZEN analysis tools, only in cells where MTs remain stable in focus during the acquisition time in the bleached zone. Photobleaching and background fluorescence were quantified from an equivalent zone localized opposite to the anaphase central spindle. Photobleaching and background normalization were calculated using Excel software (Microsoft) and applied to the mean fluorescence intensity values in the FRAP zone.

**Statistical analysis**. The number of cells counted per experiment for statistical analysis is indicated in figure legends. Graphs were created using Microsoft Excel or GraphPad Prism software and error bars represent the standard deviation unless otherwise specified. $P$-values were calculated using a two-tailed Student's $t$ test. $P > 0.05$ was considered as not significant and by convention $*P < 0.05$, $**P < 0.01$ and $***P < 0.001$.

**Data availability**. The authors declare that the main data supporting the findings of this study are available within the article and its Supplementary Information files.

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

## Acknowledgements

The experiments were performed within the France-BioImaging national research (ANR-10-INSB-04, "Investments for the future"), at Montpellier Ressources Imagerie facility (MRI), Montpellier. We particularly thank V. Georget, S. De Rossi and S. Lachambre for their assistance regarding microscopy and image analysis and M. Boyer for her help with cytometry. We also thank the Montpellier Genomic Collection platform (S. Fromont and F. Lionneton) and the Functional Proteomic Platform (S. Urbach and E. Demettre). We also thank G. Pazour for Flag-IFT IMCD stable cell lines, P. Wadsworth for the GFP-αtubulin LLC-PK1 cell line, the Hyman Lab for GFP-MKLP2 HeLa Kyoto cells, the Fachinetti and Holland labs for reagents used in generating the IFT88-AID targeted cell line, D. Fesquet for MKLP1 antibody, M. Mishima for Phospho S708 MKLP1 antibody and the members of the team for discussions and critical reading of the manuscript. This work was supported by the ANR "Chaire d'excellence" CilMitoCyst (ANR-12-CHEX-005 to B.D.), the Marie Curie career integration grant (CilMitoPatho to B.D.), the Fondation pour la Recherche Médicale (Partenariat Fondation Schlumberger pour l'Education et la Recherche to B.D.), the Fondation ARC pour la Recherche sur le Cancer (B.D.) and the CNRS (B.V., C.A., B.D.). Work in the Echard lab is supported by the Institut Pasteur, CNRS, INCa and ANR (AbsCyStem and Cytosign).

## Author contributions

N.T. conceived, designed and executed the experimental work. N.T. interpreted the data, assembled the figures and wrote the manuscript. B.V.: Carried out and analyzed the FRAP experiments, generated the AID cell lines and helped with manuscript editing. C.A. and A.D.: Helped technically with experiments. M.R. performed the two-hybrid experiments and A.E. provided tools, advices and helped with manuscript editing. M.T. and E.L.: Provided recombinant IFT proteins. B.D.: Supervised the project, designed and analyzed the experimental work, edited the figures and wrote the manuscript.

## Additional information

**Competing interests:** The authors declare no competing financial interests.

