## [Peer Review File · Nature Communications]

Reviewers' Comments:

Reviewer #1 (Remarks to the Author)

The authors report an unexpected role for intraflagellar transport (IFT) proteins in cytokinesis. IFT88 and IFT 27, components of the IFT-B complex, are needed for proper organization of the mitotic central spindle. The authors go on to dissect the molecular interactions and mechanisms by which the IFT proteins, together with other mitotic and spindle proteins such as MKLP1, MgcRacGAP and Aurora B, spatially control symmetrical cleavage furrow ingression. Using kidney epithelial cells that in 3D culture form multicellular structure with lumens, they show that depletion of IFT88 causes formation of multiple lumens, rather than the more typical single lumen.

Overall, this is apparently an original and potentially interesting result. The work apparently is well done, though I am not an expert on mitosis. It is well presented.

My chief concern is that the data is entirely in cell culture. With 3D kidney epithelial cell culture there are numerous perturbations that cause multiple lumens and so the physiological relevance of the multiple lumen phenotype is unknown. See for instance PMID 18349078. I would find this study far more appealing if it included some in vivo data, such as loss of an IFT protein during an early stage of kidney tubule lumen formation. See for instance PMID 23487309. If the authors' conclusion applies in real life, there should be some sort of defect in lumen formation in vivo.

The authors speculate on the relevance to polycystic kidney disease. This disease is not a problem with multiple lumens, but rather with lumens that are greatly expanded.

Reviewer #2 (Remarks to the Author)

Talet et al identify a function of IFT proteins (IFT88 and IFT27) in cytokinesis. They show that depletion of IFT proteins cause asymmetric ingression of the cleavage furrow, leading to mispositioning of the lumen initiation site of renal epithelial cells in 3D cultures. Following a meticulous and elegant approach, they show that IFT proteins regulate the positioning of Aurora B through MKLP2 at the mitotic spindle midzone, which is responsible for the subsequent recruitment of MKLP1 and MgcRacGAP at the same place to initiate cleavage furrow formation. Cells depleted of IFT88 or IFT27 show defects in concentrating AurkB and MKLP2 at the midzone. Consistently, MKLP1 and MgcRacGAP do not accumulate properly at the midzone leading to asymmetric distribution of RhoA and cleavage furrow ingression. The authors used all appropriate controls and state-of-the-art tools to substantiate their claims. This work is significant because it suggests that cyst formation seen in animal models lacking IFT proteins may be caused by defects in cytokinesis in addition to or rather than ciliary defects. The evidence implicating IFT proteins in extraciliary functions is steadily increasing and may change the way we think about the role of cilia in kidney cysts. In my view the manuscript provides novel information, data are of high quality, and the conclusions are supported. Therefore, I recommend it for publication in Nature Communications. However, additional experiments should be considered to further strengthen conclusions.

1. Protein-protein interactions were mostly done in cells with stable overexpression of tagged proteins. Key protein-protein interactions, such as the IFT88-Aurora B kinase in cells in anaphase should be shown using the endogenous proteins.
2. MKLP1 (Kif23) and MKLP2 (Kif20A) may have effects on cilia. The authors may want to test whether they are present in cilia and if so, whether their depletion can affect cilia formation.
3. Data using 3D cultures still cannot rule out the possibility that IFT88 mediates its effects on lumen positioning through effects on cilia. If cilia are not critical in lumen initiation, the authors

should be able to use an epithelial cell line lacking primary cilia (perhaps, HCT116 or DLD-1) and show that lumen positioning is normal unless IFT88 (or IFT27), but also MKLP1, MgcRacGAP, MKLP2, or Aurora B is depleted. Depletion of PRC1 should not have the same effect. If the authors can come up with a way to mislocalize, instead of depleting bona fide regulators of cytokinesis, it would be ideal.

4. It would be interesting to show that the Auxin-induced degradation of IFT88 can also cause a delay in the relocalization of Aurora B kinase in ciliated cells, such as the RPE1 cells. An RPE1 line expressing TIR9a is available. The HCT116 cells may not be ciliated. If they are, no need to do this experiment.

Reviewer #3 (Remarks to the Author)

In this study Taulet et al investigate the role of intraflagellar transport (IFT) proteins on the ingression of the cytokinetic furrow after anaphase onset. They show that siRNA-mediated depletion of IFT88 and IFT 27 affects the assembly of the central spindle and impairs the localization of some central spindle components, including the two members of the centralspindlin complex (MKLP1 and MgcRacGAP), Aurora B, and MKLP2 during anaphase. However, central spindle assembly and the localization of these central spindle components become normal at later stages and cytokinesis is successful, albeit the authors report an increase in the number of cells showing asymmetric (i.e. unilateral) ingression of the cleavage furrow. This asymmetric furrow ingression seems to affect lumen formation when kidney cells form cysts on a 3D matrix. In an attempt to find a mechanism that could explain how IFT proteins could regulate the assembly of the central spindle, the authors identified the proteins interacting with IFT27 in metaphase cells using immunoprecipitation (IP) followed by mass spectrometry (MS). This approach led to the identification of Aurora B, the kinase component of the chromosomal passenger complex (CPC), and the authors tried to confirm this interaction using both reciprocal immuno-precipitations and proximity ligation assays (PLA). From these results, the authors propose that IFT proteins control furrow formation and central spindle assembly by binding to the MKLP2/Aurora B complex, which in turn regulates the localization of centralspindlin and RhoA activation at the cortex.

I found the data showing that IFT proteins are required for central spindle assembly and the localization of some its components convincing, but unfortunately the experiments shown to demonstrate the interaction of IFT proteins with the CPC and/or MKLP2 are, in my opinion, totally unpersuasive and either not well controlled or flawed (see comments below). In addition, the authors failed to prove that these interactions are direct.

As IFT proteins have already been described to regulate microtubule dynamics, it is very likely that defects in the localization of central spindle proteins in IFT depleted cells could simply be a secondary effect of a weak and disorganised central spindle, rather than IFT proteins playing a direct role in controlling their localization/recruitment. It is also clear that IFT proteins are required only during early anaphase, as in their absence the central spindle become robust during telophase and completion of cytokinesis is unaffected. I therefore wonder if these defects may somehow be linked to the roles of IFT proteins in earlier stages of mitosis described by the senior author in a previous study.

In conclusion, it is my opinion that without stronger, more convincing mechanistic evidence supporting the role of IFT proteins in regulating the localization and activity of MKLP2 and the CPC, this study is too descriptive and preliminary to warrant publication in Nature Communications.

Major comments:

1. The localization of IFT88 and IFT27 during cell division should be presented, with an emphasis on its distribution during anaphase and telophase. I am aware that some IFT proteins were found to localize to centrosomes and midbodies, but I don't think that their localization during anaphase and telophase has been described in detail. The authors only show a very uninformative image of

the localization of mCherry-IFT27 in Fig 1f, where IFT27 appears to be uniformly distributed throughout the cytoplasm.

2. The identification of Flag-IFT27 interactors by IP/MS is extremely poor. The main problem is that there is no control experiment (i.e. extracts from cells expressing Flag alone) to eliminate contaminants. Second, the authors did not show the localization and expression level of the Flag-IFT27 protein. Moreover, I can't understand why the authors used metaphase extracts when the aim was to identify interactors during anaphase. No detailed information was given in the M&M section on how the purification was carried out besides that: "proteins were incubated with Flag M2 beads (agarose, magnetic?) for 3 hours at 4C followed by a 3x Flag peptide elution". There is no mention of what solutions were used and if and how many washes were carried out to remove non-specific interactors. Finally, the authors failed to provide a full list of interactors and only mentioned in the text that Aurora B was identified, but they did not report the protein coverage and/or the number of peptides identified. Were the other CPC components and MKLP2 identified in this experiment?

3. The complementary IP experiments to confirm the interactions shown in Fig 2a suffer from similar problems. Why were metaphase extracts instead of anaphase extracts used? It is unclear what the controls are in Fig. 2a, but it does not look like the authors used the proper controls, i.e. extracts from cells expressing the tags (myc or Flag) alone.

4. The PLA experiments in Fig. 2c also lack proper controls (i.e. Aurora B and IFT88 antibodies separately). Furthermore, PLA dots appear scattered randomly; they do not reflect Aurora B localization and often do not overlap with microtubules. These experiments are not convincing and a more straightforward biochemical evidence of a direct interaction between IFT proteins and the CPC/MKLP2 should be provided.

5. Why there is tubulin staining in the middle panel of Fig. 1a? There should be no tubulin antibody in this control. Moreover, the strongest PLA dots are far away from the spindle. I have the distinct impression that the authors have not mastered this technique.

6. I am seriously worried by the high percentage (20-30%) of defects observed in several siRNA controls (Fig. 1h, 2e, 3f, and 4b). What is the reason for this? For example, how can it be possible that more than 30% of control cells show mislocalized MKLP2 (Fig. 3f). Are the authors sure that their control siRNA has no off-target effects?

7. The authors used IFT88 siRNA in many of their experiments, but never demonstrated the specificity of their siRNA using an RNAi rescue construct as shown for IFT27 in Fig. 1f.

Minor:

1. Abstract, third line from the top: ".. and requires the recruitment and activation at this site of ..". It is unclear what "this site" refers to.

2. The introduction is too succinct and does not provide sufficient information to readers.

3. Page 4, the sentence: "Given that MKLP1 and MgcRacGAP localize to the central spindle in an Aurora B dependent manner" is incorrect and the references are wrong. Aurora B promotes the clustering of centralspindlin by phosphorylating MKLP1 (PMID: 20451386); there is no published evidence that Aurora B-mediated phosphorylation of MgcRacGAP is important for centralspindlin localization to the central spindle.

4. Page 6: the authors can't claim that asymmetric furrow ingression was "associated" with asymmetric RhoA localization without the support of time-lapse experiments.

5. Panels e-h in Fig. 4 are not conceptually linked with the rest of the data and the experiments were carried out in a completely different system. They should be in a separate figure.

6. Some of the authors' statements are too strong (e.g. this distribution was dramatically affected - page 6, third line from the top). I would encourage them to be more honest by using less exaggerated attributes and to limit the use of the verb "demonstrate" throughout the paper.

Response to the reviewers' comments

We thank the three Reviewers for their useful comments and for suggested experiments. We have provided additional experiments and Figures as well as explanations in the revised manuscript to address all their comments.

New experimental data have been added as. Fig. 1d; Fig. 2a,b,e; Fig. 3 a,b; Fig. 4e,f; Fig. 5b, c ; Supplementary Fig. S1a, b; S4; S5; Supplementary. Table 1

We provide below a full response to each comment raised by the Reviewers. We believe that the added experimental data and Reviewer's suggestions helped us to reinforce the conclusions of the manuscript.

Reviewer 1

Overall, this is apparently an original and potentially interesting result. The work apparently is well done, though I am not an expert on mitosis. It is well presented.

We would like to thank the reviewer for his/her time in assessing our work and for his/her comments.

My chief concern is that the data is entirely in cell culture. (...) I would find this study far more appealing if it included some in vivo data, such as loss of an IFT protein during an early stage of kidney tubule lumen formation. See for instance PMID 23487309. If the authors' conclusion applies in real life, there should be some sort of defect in lumen formation in vivo.

Previous works in IFT88 mutant mice reported kidney disease and defects in kidney tubule formation (PMID:8191288 ; PMID:10804177). Initial multifocal microscopic dilation of proximal tubules was followed rapidly by marked dilations and cyst formation (PMID:8191288). However, the reason why IFT88 depletion led to this phenotype was unknown.

In this manuscript, our goal is to provide molecular and cellular mechanistic explanations for the observed initial defects in kidney tubule formation observed in animal models. Using cutting edge cell biology techniques, 2D and 3D cell culture models, we were able to recapitulate the *in vivo* defects (see also below) and explain the initial defects observed *in vivo*. This is now clearly explained in the main text (pages 2-3; page 8; page 10).

In addition, to better highlight the relevance of our results with previous *in vivo* studies, we have now included in the introduction and the discussion details regarding two key publications linking IFT88 (also referred to as Tg737 gene, orpk mouse) to polycystic kidney disease and describing the initial defects in lumen formation observed *in vivo*:

-Moyer et al Science 1994 : "lesions in the kidneys and livers of TgN737Rpw mutant mice were markedly similar to those seen in human ARPKD" .. "In the kidneys, an initial mild, multifocal, microscopic dilation of the proximal tubules was followed rapidly by marked dilation and cyst formation of the collecting tubules."

-Murcia et al Development 2000 : "Tg737D2-3bGal/orpk heterozygotes (...) recapitulate the entire phenotype of orpk homozygous mutants (...). These lesions include (...) proximal tubule dilations and collecting duct cysts in the kidney."

The authors speculate on the relevance to polycystic kidney disease. This disease is not a problem with multiple lumens, but rather with lumens that are greatly expanded.

We agree with the fact that lumens are generally greatly expanded in polycystic kidney disease. However, polycystic kidneys and tubular defects observed *in vivo* in IFT88 mutant mice actually derive from an initially mild, multifocal and microscopic dilation of the proximal tubules. In PMID:8191288, Moyer et al 1994 indeed noticed that “In the kidneys, an initial mild, multifocal, microscopic dilation of the proximal tubules was followed rapidly by marked dilation and cyst formation of the collecting tubules.”.

Upon IFT88 depletion, we precisely observe multifocal lumens in 3D cultures (at 3-6 cells stage, Supplementary Fig. S6; page 8). Thus, our results in cell culture recapitulate what was initially observed *in vivo* when kidney tubule defects first appear before observing a more dramatic enlargement. Importantly, our work now provides a cellular and molecular mechanism for the contribution of cytokinesis defects to the initial multifocal kidney defects observed in IFT88 mutant mice. The complementarity between our mechanistic results in 3D cultures and previous *in vivo* work in IFT88 mutant mice is now explicitly commented in the revised version of the manuscript (page 10).

Reviewer 2

The authors used all appropriate controls and state-of-the art tools to substantiate their claims. This work is significant because it suggests that cyst formation seen in animal models lacking IFT proteins may be caused by defects in cytokinesis in addition to or rather than ciliary defects. The evidence implicating IFT proteins in extraciliary functions is steadily increasing and may change the way we think about the role of cilia in kidney cysts. In my view the manuscript provides novel information, data are of high quality, and the conclusions are supported. Therefore, I recommend it for publication in Nature Communications. However, additional experiments should be considered to further strengthen conclusions.

We would like to thank the reviewer for his/her careful reading of the manuscript and for his/her suggestions to perform additional experiments that definitely strengthened our conclusions. As requested by the reviewer, two key questions were addressed:

a) To strengthen IFT/MKLP2 interaction (point 1), we now provide two-hybrid data indicating a direct interaction between IFT88 and MKLP2. This approach also led to the characterization of the IFT88 binding domain of MKLP2. Importantly, we confirmed the direct interaction between IFTs and MKLP2 biochemically using recombinant proteins. Two-hybrid and biochemical results are now provided in Fig. 3a and b and Supplementary Fig. S4b. We also provide novel co-immunoprecipitation experiments using a HeLa BAC line expressing GFP-MKLP2 under the control of its endogenous promotor (Supplementary Fig. S4a).

b) To address the ciliary / cytokinesis questions (raised in points 2, 3 and 4), we now provide novel data using low dose of MKLP2 inhibitor that delays Aurora B relocalization (Supp Fig S4d). We show that treatment with the MKLP2 inhibitor phenocopies the effects of IFT88 depletion on both cleavage furrow ingression and lumen positioning. Data are now included in the main figures (Fig. 4e and f and Fig. 5b and c) and definitely strengthen our point. Importantly, at the same dose of inhibitor, no effect was observed on cilia formation. These data are now provided in Supplementary Fig. S5b and c.

More detailed answers to each specific point are provided below:

1. Protein-protein interactions were mostly done in cells with stable overexpression of tagged proteins. Key protein-protein interactions, such as the IFT88-Aurora B kinase in cells in anaphase should be shown using the endogenous proteins.

As requested, we have now improved key protein-protein interactions.

- The available antibodies working for immunoprecipitation did not allow to detect endogenous interactions. However, we now provide novel co-immunoprecipitation experiments using a HeLa BAC line expressing GFP-MKLP2 under the control of its endogenous promotor (Supplementary Fig. S4a). GFP-TrapA beads were used for co-IPs of GFP-tagged MKLP2 expressed at endogenous levels. These IPs were performed on cells enriched in anaphase (80% of mitotic cells including an average of 50% of anaphase). This is now clearly mentioned in the methods.

-To demonstrate direct protein-protein interactions, we now provide additional data using both two hybrid assays and *in vitro* biochemical experiments with recombinant proteins. More specifically, we now show that IFT88 directly interacts with MKLP2 by two-hybrid and mapped the binding domain.

We next confirmed that this domain directly interacts *in vitro* with IFTs using recombinant proteins. These key results have been included in Fig. 3a and b and Supplementary Fig. S4b.

2. MKLP1 (Kif23) and MKLP2 (Kif20A) may have effects on cilia. The authors may want to test whether they are present in cilia and if so, whether their depletion can affect cilia formation.

Since we describe here a role for IFTs on the transport of MKLP2/Aurora B in mitosis and a direct interaction between IFT88 and MKLP2, we agree with the reviewer that testing if the MKLP2/Aurora B complex could also localize to cilia and affect cilia formation was an important point to control.

In the revised version of the manuscript, we show that:

- Aurora B is not detected at the cilium of LLC-PK1 cells (new supplementary Fig. S5a) indicating that Aurora B should not impact on ciliogenesis and that the AuroraB/MKLP2 complex is not localized at the cilium.

- MKLP2 is detected at the centrosome of interphase cells and at the base of the cilium but not in the cilium (new supplementary Fig. S5a). As requested, we therefore used paprotrain, a MKLP2 inhibitor, to test whether MKLP2 inhibition affects cilia formation. A dose of MKLP2 inhibitor (5 μ M) that delays Aurora B relocalization, phenocopies cleavage furrow defects and perturbs lumen positioning as observed upon IFT88 depletion (see below comment 3: reasons for using inhibitor instead of depletion). In these conditions, MKLP2 inhibition does not affect cilia formation in RPE cells (new supplementary Fig S5b and S5c). A dynein inhibitor known to disrupt ciliogenesis (Ciliobrevin D), was used as positive control.

- MKLP1 was shown previously to localize to the basal body (PMID:21246755). We did not further investigate the role of MKLP1 in cilia formation since we now show a direct interaction between IFT88 and MKLP2.

3. Data using 3D cultures still cannot rule out the possibility that IFT88 mediates its effects on lumen positioning through effects on cilia. If cilia are not critical in lumen initiation, the authors should be able to use an epithelial cell line lacking primary cilia (perhaps, HCT116 or DLD-1) and show that lumen positioning is normal unless IFT88 (or IFT27), but also MKLP1, MgcRacGAP, MKLP2, or Aurora B is depleted. Depletion of PRC1 should not have the same effect. If the authors can come up with a way to mislocalize, instead of depleting bona fide regulators of cytokinesis, it would be ideal.

To address the reviewer's comments regarding the potential role of cilia on lumen positioning, we now provide 3 additional key results:

- As mentioned by the reviewer, depletion of cytokinesis regulators cannot be used in this case since the effect would be too dramatic (cytokinesis failure) to monitor any downstream effect. To mislocalize a bona fide cytokinesis regulator, we used a low dose of the MKLP2 inhibitor to delay Aurora B relocalization towards the central spindle. In these conditions, we found that abnormal Aurora B localization phenocopies the effect of IFT88 depletion both on cytokinesis (asymmetric ingression) and lumen positioning (new Fig4e, f and Fig.5b,c). Given that no defects in cilia formation were observed using the same dose of inhibitor, these results strengthen the fact that lumen defects result from cytokinesis defects.

- As a second approach, we monitored the presence of cilia in two cells stage 3D structures. Very low number (maximum 15%) of cilia were observed and no correlation was found between the presence of cilia (Arl-13 staining) and lumen initiation site positioning. In addition, MKLP2 inhibition did not

change the overall number of cilia whereas lumen mispositioning was strongly increased (>3 fold). This result again strengthens the fact that the mispositioning of the lumen initiation site upon MKLP2 inhibition is not due to an absence of cilia.

- Finally, as requested by the reviewer, we also used DLD1 cells to monitor lumen positioning in cells lacking primary cilia. We confirmed the absence of cilia in these cells (Arl-13 staining) in 2 cell stage 3D cultures, as previously described (PMID:21602792). Actin patches were centrally localized in most (approximately 70%) DLD1 3D structures. As anticipated, this central localization was perturbed upon auxin treatment that induced IFT88 degradation. This result shows that lumen initiation site is central in cells lacking cilia, unless IFT88 is depleted. Given that DLD1 cells are not, in our opinion, as good as LLC-PK1 for monitoring lumen formation, we decided not to include the figure in the paper (see below), unless the reviewer requests it.

In DLD1 cells, lumen initiation site positioning is perturbed upon auxin treatment that induced IFT88 degradation.

DLD-1 cells were grown in 3D on disk-shape micropatterns and IFT88 depletion was induced by addition of Auxin.

(A) Lumen precursor positioning (yellow arrow) was monitored at 2-cells stage, after the first cell division, using actin (green). DAPI, blue and α tubulin, red are shown.

(B) Quantification of the percentage of 2-cells stage 3D cultures with mispositioned lumen initiation site as shown by both lateral actin accumulation and nuclei mispositioning. Mean \pm S.E.M. n> 50 from two independent experiments.

4. It would be interesting to show that the Auxin-induced degradation of IFT88 can also cause a delay in the relocalization of Aurora B kinase in ciliated cells, such as the RPE1 cells. An RPE1 line expressing TIR9a is available. The HCT116 cells may not be ciliated. If they are, no need to do this experiment.

To address the reviewer's comment, we looked at cilia in HCT116 cells and they are not ciliated. We thus tried to generate either RPE or LLC-PK1 cells able to degrade IFT88 in an auxin-inducible manner. Unfortunately, we were not able to target endogenous IFT88 on both alleles in these two cell types. Therefore, we could not perform the requested experiment and hope that the siRNA data provided in ciliated LLC-PK1 cells are convincing enough for the reviewer.

Reviewer #3 (Remarks to the Author):

*I found the data showing that IFT proteins are required for central spindle assembly and the localization of some its components convincing, but unfortunately the experiments shown to demonstrate the **interaction of IFT proteins with the CPC and/or MKLP2** are, in my opinion, totally unpersuasive and either not well controlled or flawed (see comments below). In addition, the authors failed to prove that these **interactions are direct**.*

We would like to thank the reviewer for his/her time in assessing our work and for his/her suggestions to strengthen our paper regarding direct interactions data.

The additional results obtained during the revision process definitely improved our paper since we now provide evidence for a direct interaction between IFT88 and MKLP2. More specifically, we now show that IFT88 directly interacts with MKLP2 by two-hybrid assays and mapped the binding domain. We next provide biochemical evidence confirming that this domain directly interacts with IFTs *in vitro* using recombinant proteins. These key results have been included in new Fig. 3a and b and Supplementary Fig. S4b. These data provide convincing mechanistic evidence for direct interactions between MKLP2 and IFTs to ensure proper central spindle organization.

Importantly, we also provide functional evidence that IFT proteins and MKLP2 can function together since low dose of MKLP2 inhibitor, that delays Aurora B relocalization, phenocopies the asymmetric cleavage furrow ingression and lumen defects observed in IFT88 depleted cells (Fig. 4e and f; Fig. 5b and c).

We also provide in the revised manuscript all the requested controls either in the main figures or in supplementary data, and all the requested methods modifications (see below for details).

As IFT proteins have already been described to regulate microtubule dynamics, it is very likely that defects in the localization of central spindle proteins in IFT depleted cells could simply be a secondary effect of a weak and disorganised central spindle, rather than IFT proteins playing a direct role in controlling their localization/recruitment. I therefore wonder if these defects may somehow be linked to the roles of IFT proteins in earlier stages of mitosis described by the senior author in a previous study.

To prove that the observed anaphase defects are not due to earlier mitotic defects, we took advantage of the monopolar cytokinesis approach and combined it to the Auxin inducible degron technique. This approach ensures an accurate depletion of IFT88 when cells are already synchronized, ruling out any contribution of earlier defects to the delay in Aurora B relocalization observed (Fig. 2j and Fig. S3).

Major comments:

1. The localization of IFT88 and IFT27 during cell division should be presented, with an emphasis on its distribution during anaphase and telophase. I am aware that some IFT proteins were found to localize to centrosomes and midbodies, but I don't think that their localization during anaphase and telophase has been described in detail. The authors only show a very uninformative image of the localization of mCherry-IFT27 in Fig 1f, where IFT27 appears to be uniformly distributed throughout the cytoplasm.

Immunofluorescence images of endogenous IFT88 in LLC-PK1 cells are now provided in supplementary Fig. S1a and show endogenous IFT88 puncta along the length of central spindle MTs in anaphase. Proximity Ligation Assay images of IFT88 associated with microtubules in prometaphase, metaphase and anaphase are also shown in new Fig S1b.

Note that the image presented in Fig. 1f was not provided to precisely localize IFT27 but only to compare cells expressing or not mCherry-IFT27 in the rescue experiments. Unfortunately, we did not find an antibody working in immunofluorescence in pig LLC-PK1 cells for IFT27. However, we now provide a detailed analysis of the localization of one of the IFT components (IFT88), as mentioned above.

2. The identification of Flag-IFT27 interactors by IP/MS is extremely poor. The main problem is that there is no control experiment (i.e. extracts from cells expressing Flag alone) to eliminate contaminants.

IMCD cells were used as control for MS to eliminate contaminants and the selected candidate interactor Aurora B is now validated using Flag alone as negative control (see new Fig. 2a).

Second, the authors did not show the localization and expression level of the Flag-IFT27 protein.

Levels of Flag-IFTs in IMCD cells, including Flag-IFT27, are now specifically shown with red asterisks in Supplementary Fig. S2b.

Moreover, I can't understand why the authors used metaphase extracts when the aim was to identify interactors during anaphase. No detailed information was given in the M&M section on how the purification was carried out besides that: "proteins were incubated with Flag M2 beads (agarose, magnetic?) for 3 hours at 4C followed by a 3x Flag peptide elution". There is no mention of what solutions were used and if and how many washes were carried out to remove non-specific interactors.

The aim of the proteomic approach was to identify novel mitotic interactors. We thus used cells arrested after conventional mitotic block. This is now clearly mentioned in the text: "we performed an unbiased mass-spectrometry approach on kidney cells enriched in mitosis to identify novel IFT mitotic interacting partners". Validations were then done for anaphase interactors using cells enriched in anaphase (see new methods section).

As requested, the Methods section has been improved and additional information regarding procedures, washes and solutions have been included.

Finally, the authors failed to provide a full list of interactors and only mentioned in the text that Aurora B was identified, but they did not report the protein coverage and/or the number of peptides identified. Were the other CPC components and MKLP2 identified in this experiment?

The full list of interactors identified in this study is now provided as supplementary .xls files. Protein coverage and number of peptides are provided in the table. Several cytokinesis regulators were detected as IFT27 interacting partners including Aurora B and the centralspindlin complex MKLP1 and MgcRacGAP. Aurora B was further validated as shown in Fig2a and b. Other CPC components and MKLP2 were not identified under these conditions.

3. The complementary IP experiments to confirm the interactions shown in Fig 2a suffer from similar problems. Why were metaphase extracts instead of anaphase extracts used? It is unclear what the controls are in Fig. 2a, but it does not look like the authors used the proper controls, i.e. extracts from cells expressing the tags (myc or Flag) alone.

Extracts of cells enriched in anaphase were actually used. Details regarding synchronization protocols have now been provided in methods: "For immunoprecipitation experiments, HeLa Kyoto cells were synchronized in mitosis using nocodazole 100 ng/ml for 15h followed by 1h release or double thymidine (2 mM) block followed by 9h release to reach 80% of mitotic cells including an average of 50% of anaphase."

Fig. 2a and b now include controls with flag and/or myc alone, as requested by the reviewer. Fig. S4a also includes GFP alone as control. This is now written in detail in the Methods section.

4. The PLA experiments in Fig. 2c also lack proper controls (i.e. Aurora B and IFT88 antibodies separately). Furthermore, PLA dots appear scattered randomly; they do not reflect Aurora B localization and often do not overlap with microtubules. These experiments are not convincing and a more straightforward biochemical evidence of a direct interaction between IFT proteins and the CPC/MKLP2 should be provided.

Regarding the PLA presented in Fig. 2c, we had all the requested controls in the initial version of the paper but did not show them in all figures. IFT88 alone was already shown in Fig. 1a and Aurora B in Fig. S4c. We also provided in Fig. 2c siRNA IFT88 showing a decrease in PLA dots for IFT88/Aurora B upon IFT88 depletion as a control. To fully address the reviewer's comments, we now refer to the corresponding controls in the figure legends.

As requested, we now provide more straightforward experiments to demonstrate direct interactions between MKLP2 and IFTs. In the revised version of the manuscript, we show that IFT88 directly interacts with MKLP2 by two-hybrid assays and mapped the binding domain. We next provide biochemical evidence confirming that this domain directly interacts with IFTs *in vitro* using recombinant proteins. These key results have been included in new Fig. 3a and b and Supplementary Fig. S4b.

5. Why there is tubulin staining in the middle panel of Fig. 1a? There should be no tubulin antibody in this control. Moreover, the strongest PLA dots are far away from the spindle. I have the distinct impression that the authors have not mastered this technique.

The α -tubulin staining seen in Fig. 1a corresponds to endogenous α -tubulin, which is visualized in the cells using an α tubulin-FITC antibody after performing the PLA. This middle panel is used to control for mitotic stages and localize PLA dots relative to central spindle MTs. Details in this procedure can be found in the Methods section. The big red PLA aggregates do not correspond to the average size of PLA dots and cannot be considered as specific PLA staining.

6. I am seriously worried by the high percentage (20-30%) of defects observed in several siRNA controls (Fig. 1h, 2e, 3f, and 4b). What is the reason for this? For example, how can it be possible than more than 30% of control cells show mislocalized MKLP2 (Fig. 3f). Are the author sure that their control siRNA has no off-target effects?

To address the reviewer's concerns regarding the percentage of defects in control siRNA, we now provide the quantification for abnormal MTs organization (new Fig 1d) and for Aurora B mislocalization (new Fig 2e) in non-treated conditions. No significant difference between non-treated and control siRNA conditions were found, confirming the absence of off-target effects for control siRNAs. This is now mentioned in the corresponding figure legends.

7. The authors used IFT88 siRNA in many of their experiments, but never demonstrated the specificity of their siRNA using an RNAi rescue construct as shown for IFT27 in Fig. 1f.

We tried to rescue the anaphase defects by overexpressing IFT88 in IFT88 depleted cells. However, as previously published (PMID:17264151), the intraflagellar transport component IFT88/polaris is a centrosomal protein regulating G1-S transition in non-ciliated cells. Thus, IFT88 overexpression interferes with the cell cycle. In IFT88 overexpressing cells, the number of anaphase observed under

these conditions was therefore too low. This was not the case for IFT27, for which we provide the rescue experiments.

To overcome this problem and strengthen our results regarding the specificity of the phenotype observed upon IFT88 depletion, we used 2 independent siRNA sequences in independent cell lines (details are provided in the Methods section). In addition, we used an Auxin inducible degron approach for IFT88 that gave similar phenotype. In this method, the AID tag was fused to the endogenous alleles of IFT88 using CRISPR/CAS9. IFT88 degradation was then specifically induced upon auxin treatment. This provides a specific and independent way to deplete IFT88, without using siRNAs.

Minor:

1. Abstract, third line from the top: “.. and requires the recruitment and activation at this site of..”. It is unclear what “this site” refers to.

This has been modified.

2. The introduction is too succinct and does not provide sufficient information to readers.

This has been modified and a longer introduction is now provided (the paper was initially written as a letter).

3. Page 4, the sentence: “Given that MKLP1 and MgcRacGAP localize to the central spindle in an Aurora B dependent manner” is incorrect and the references are wrong. Aurora B promote the clustering of centralspindlin by phosphorylating MKLP1 (PMID: 20451386); there is no published evidence that Aurora B-mediated phosphorylation of MgcRacGAP is important for centralspindlin localization to the central spindle.

The sentence has been modified according to the reviewer’s suggestion and the reference mentioned is included in the text.

New version: “In anaphase, Aurora B is translocated to the central spindle by the kinesin MKLP2 and promotes the clustering and accumulation of the MKLP1/MgcRacGAP complex (centralspindlin), by phosphorylating MKLP1.”

4. Page 6: the authors can’t claim that asymmetric furrow ingression was “associated’ with asymmetric RhoA localization without the support of time-lapse experiments.

This has been modified.

5. Panels e-h in Fig. 4 are not conceptually linked with the rest of the data and the experiments were carried out in a completely different system. They should be in a separate figure.

As requested, the initial Fig. 4 has been split into 2 figures (Fig. 4 and Fig. 5). This also allowed the addition of the MKLP2 inhibitor data requested by reviewer 2 on both Fig. 4 and Fig. 5. The model is now presented in Fig. 6.

6. Some of the authors’ statements are too strong (e.g. this distribution was dramatically affected - page 6, third lane from the top). I would encourage them to be more honest by using less exaggerated attributes and to limit the use of the verb “demonstrate” throughout the paper.

The text has been modified according to the reviewer’s suggestions to remove “dramatically” or “demonstrate”.

Reviewers' Comments:

Reviewer #1:

Remarks to the Author:

I am satisfied with the authors' response. I think this paper is definitely worth publishing in Nat Comms.

My apologies for taking so long.

Reviewer #2:

Remarks to the Author:

The authors have done a great job in providing new data to support conclusions. The strength of the paper is the identification of a previously unrecognized role of IFT proteins in cytokinesis supported by meticulous work the authors have done to support their claim. The manuscript is addressing a strictly cell biological question and the approach taken to address this question rigorously in cell culture is appropriate. I am happy to recommend the manuscript for publication in Nature Communications.

Reviewer #3:

Remarks to the Author:

This revised version of the manuscript of Delaval and co-workers has been improved by the additional results, in particular by the experiments supporting a direct interaction between IFT proteins and MKLP2. It is really a pity, however, that the authors decided not to follow my suggestion to characterise the IFT interactome during anaphase/telophase. From their synchronization protocol it is clear that, after 1h release from RO3306, cells were enriched in metaphase. The lack of information about IFT interactors in anaphase considerably weakens their study. For example, it is very difficult to rationalize why the authors decided to investigate the interaction between IFT proteins and Aurora B/MKLP2 when their mass spectrometry data indicated a very weak association with Aurora B – just one peptide was identified without any other CPC component– and no association with MKLP2. By contrast, the two components of the centralspindlin complex, MKLP1/KIF23 and RacGAP1 were more abundant and yet the authors did not investigate whether IFT proteins interacted with this complex. This would have been much a much more straightforward story, considering that centralspindlin is a well-known central spindle microtubule bundling factor and its accumulation is reduced after IFT siRNA.

There are still some issues that, in my opinion, should be addressed before publication.

1. The authors did not address whether IFT proteins regulate the localization and dynamics of the centralspindlin complex (see above) and therefore this possibility cannot be excluded and should be discussed.
2. To convincingly demonstrate that depletion of IFT proteins affects centralspindlin phosphorylation by Aurora B the authors should check the level of MKLP1 phosphorylation at S708 (see Douglas et al. 2010, PMID: 20451386).
3. The authors cannot claim that Aurora B translocation to the central spindle was “delayed” after IFT depletion (Fig 2) without the support of time-lapse studies.
4. The authors should explain why about 30% of control cells show accumulation of Aurora B at chromosomes in Fig. 2j whereas no Aurora B at chromosomes was observed in the control cells of the siRNA experiments shown in Fig. 2i.
5. The authors should confirm that the localization of IFT88 to the spindle (Fig. S1a) disappears after IFT88 depletion.
6. The decrease of MKLP1 and MgcRacGAP (Fig. 1g) should be quantified.
7. The in vitro pull down assay does not indicate a strong interaction between IFT proteins and

MKLP2 (Fig. 3b and S4b) and the authors do not show serial dilution in the Y2H (Fig. 2a). Therefore their statement that "a strong interaction was detected between IFT88 and MKLP2" (page 6) is unsubstantiated and should be eliminated.

8. How were recombinant IFT proteins expressed and purified (Fig 3b and S4b)? This information must be included.

9. The signals relative to the Flag and Myc tags should be shown in Figs 2a and b.

10. Collection of images at 5 min intervals is insufficient to properly analyze MKLP2-GFP dynamics after IFT88 siRNA (Fig 3f and relative movie).

Response to the reviewers' comments

We would like to thank again the reviewers for their constructive comments and their input during the revision process that definitely strengthened our conclusions. We are pleased to see that reviewers 1 and 2 were convinced by the new data provided during the revision process and that they raised no further concerns.

Regarding reviewer 3 concerns (revision round 2), we provide, as requested, additional experiments as well as modifications in the revised version of the manuscript to address his comments.

New experimental data have been added as Fig 1b, Supplementary Fig 1e, Supplementary Fig 3b, Supplementary Fig 4d.

All changes made in the manuscript text file have been highlighted as requested.

We also provide below a full response to each comment raised by Reviewer 3.

REVIEWER 3

This revised version of the manuscript of Delaval and co-workers has been improved by the additional results, in particular by the experiments supporting a direct interaction between IFT proteins and MKLP2. It is really a pity, however, that the authors decided not to follow my suggestion to characterise the IFT interactome during anaphase/telophase. From their synchronization protocol it is clear that, after 1h release from RO3306, cells were enriched in metaphase. The lack of information about IFT interactors in anaphase considerably weakens their study. For example, it is very difficult to rationalize why the authors decided to investigate the interaction between IFT proteins and Aurora B/MKLP2 when their mass spectrometry data indicated a very weak association with Aurora B – just one peptide was identified without any other CPC component– and no association with MKLP2. By contrast, the two components of the centralspindlin complex, MKLP1/KIF23 and RacGAP1 were more abundant and yet the authors did not investigate whether IFT proteins interacted with this complex. This would have been much a much more straightforward story, considering that centralspindlin is a well-known central spindle microtubule bundling factor and its accumulation is reduced after IFT siRNA.

We would like to apologize for not performing the initial proteomic approach on anaphase cells but we faced technical issues to more precisely synchronize the cells. Indeed, we tried several synchronization methods on IMCD cells expressing Flag-IFTs (double thymidine block, RO3306, nocodazole) but these cells did not release from the block in a perfectly synchronous manner. This synchronization issue combined to the short time frame of anaphase in these cells did not lead to an enrichment in anaphase cells sufficient to perform mass spectrometry. Given that the aim of the proteomic approach was to identify novel mitotic candidates interacting with IFTs, we therefore chose to use RO3306 for the mass spectrometry to enrich enough cells in mitosis and then confirm the results in anaphase in other cell types easier to synchronize.

This approach successfully led to the identification of Aurora B as a potential candidate that was then characterized in depth in our paper. Characterizing the IFT interactome during anaphase/telophase would of course be interesting in the future but we feel goes beyond the scope of our manuscript at this stage of the revision process.

There are still some issues that, in my opinion, should be addressed before publication.

1. The authors did not address whether IFT proteins regulate the localization and dynamics of the centralspindlin complex (see above) and therefore this possibility cannot be excluded and should be discussed.

We agree with the reviewer that IFT proteins may also interact and/or regulate other complexes in dividing cells including the centralspindlin complex. We initially decided in this paper to focus on Aurora B since we rapidly confirmed an interaction between IFTs and Aurora B on cells enriched in anaphase. According to his previous comments (first round of revision), we further characterized this complex by identifying direct interactions between IFT88 and the motor responsible for Aurora B transport, MKLP2.

Additional roles of IFTs on the regulation of other motors or other proteins in anaphase including MKLP1 and MgcRacGAP can of course not be excluded. As requested by the reviewer, we now included a sentence (p10) to discuss this point.

2. To convincingly demonstrate that depletion of IFT proteins affects centralspindlin phosphorylation by Aurora B the authors should check the level of MKLP1 phosphorylation at S708 (see Douglas et al. 2010, PMID: 20451386).

To strengthen the fact that depletion of IFT proteins could affect MKLP1 phosphorylation by Aurora B, we now provide, as requested, new immunofluorescence experiments using a Phospho-S708-MKLP1 antibody (Gift from Masanori Mishima). As expected, Phospho-S708-MKLP1 staining is decreased at the central spindle of anaphase cells upon IFT88 depletion. Immunofluorescence images and quantifications of fluorescence intensity are now included in Supplementary Fig. 3b.

3. The authors cannot claim that Aurora B translocation to the central spindle was “delayed” after IFT depletion (Fig 2) without the support of time-lapse studies.

To address the reviewer’s comment we expressed GFP-Aurora B in LLC-PK1 cells and made a stable cell line. We then performed time-lapse experiments (1 min interval). We indeed observed a delay in GFP-Aurora B relocalization in some cells. An example of GFP-Aurora B time-lapse experiment suggesting such a delay between control and IFT88 siRNA treated cell is provided for the reviewer. However, since it is technically challenging to get quantitative data on the delay we chose to remove “delay” from the text and replace it by “defects in Aurora B relocalization”.

Figure: Images (inverted contrast) from time-lapse microscopy of mitotic LLC-PK1 cells expressing GFP-Aurora B and transfected with control (CT) or IFT88 siRNA. Time (min). Scale bar 10µm.

4. *The authors should explain why about 30% of control cells show accumulation of Aurora B at chromosomes in Fig. 2j whereas no Aurora B at chromosomes was observed in the control cells of the siRNA experiments shown in Fig. 2i.*

Cells used in fig 2i (LLC-PK1) and 2j (HCT116) are not the same cells as mentioned in the figure legend. LLC-PK1 cells release faster and in a more synchronous manner than HCT116 into monopolar cytokinesis (see Fig. 2h LLC-PK 10 min release and Supplementary Fig 3a HCT116 30 min release), explaining the observed difference. Despite this difference in timing, a defect in Aurora B relocalization is observed in both cell types.

5. *The authors should confirm that the localization of IFT88 to the spindle (Fig. S1a) disappears after IFT88 depletion.*

We apologize for not including these images in the initial version. We now provide new immunofluorescence images showing a decrease of IFT88 staining in an anaphase cell upon IFT88 depletion. An example of a control and IFT88 knocked-down anaphase cell is now provided in Fig. 1b.

6. *The decrease of MKLP1 and MgcRacGAP (Fig. 1g) should be quantified.*

We have repeated the immunofluorescence experiments to perform the requested quantifications for both MKLP1 and MgcRacGAP. The corresponding graphs are now included in Supplementary Fig 1e.

7. *The in vitro pull down assay does not indicate a strong interaction between IFT proteins and MKLP2 (Fig. 3b and S4b) and the authors do not show serial dilution in the Y2H (Fig. 2a). Therefore their statement that “a strong interaction was detected between IFT88 and MKLP2” (page 6) is unsubstantiated and should be eliminated.*

The text has been modified according to the reviewer’s suggestion.

8. *How were recombinant IFT proteins expressed and purified (Fig 3b and S4b)? This information must be included.*

The IFT88/70/52/46 complex was expressed and purified as reported previously (Taschner et al., 2014; PMID:25349261) and was provided by MT and EL as mentioned in the author’s contribution. This is now also detailed in the methods section of the revised version (P13).

9. *The signals relative to the Flag and Myc tags should be shown in Figs 2a and b.*

We are sorry but this is not technically possible. As requested by the reviewer in the first round of revision we transfected cells with the tags only (3XFlag, Myc, GFP). The size of the 3XFlag tag or Myc tag alone is too small to be visualized on the western-blot of figure 2 a and b (2.8 kDa and 1.2 kDa respectively). The GFP tag alone which is 27 kDa is shown on figure S4A.

10. *Collection of images at 5 min intervals is insufficient to properly analyze MKLP2-GFP dynamics after IFT88 siRNA (Fig 3f and relative movie).*

As requested, we have now repeated the experiments with shorter time frame (1 and 3 minutes intervals). Still frames of the movies (1 minute interval) are now provided in supplementary Fig 4d. These experiments are only performed to assess live the distribution of GFP-MKLP2 to the central spindle in anaphase and to show a global defect in the relocalization of the motor. As previously observed with 5 min interval movies, GFP-MKLP2 relocalization to the central spindle was affected upon IFT88 depletion. To further address the reviewer’s comment we also removed “dynamics” from the text. We believe that the FRAP analysis shown in figure 3g and h provide better information on GFP-MKLP2 dynamics along microtubules.

Reviewers' Comments:

Reviewer #3:

Remarks to the Author:

The authors have satisfactorily addressed most of my comments with one exception.

In their quantification of phosphorylated S708 MKLP1 after IFT88 siRNA (Fig. S3b), they did not include the levels of total MKLP1. Indeed, IFT88 siRNA causes a similar reduction in the levels of total MKLP1 ((Fig S1e) and therefore their experiment does not support the interpretation that IFT88 is necessary for Aurora B-mediated phosphorylation of MKLP1. They should either repeat the experiment properly (there are many commercial anti-MKLP1 antibodies generated in various species) or eliminate this statement from their manuscript.

Response to the reviewers' comments

We are pleased to see that reviewer 3 is satisfied with the new data provided during the revision process.

REVIEWER 3

The authors have satisfactorily addressed most of my comments with one exception. In their quantification of phosphorylated S708 MKLP1 after IFT88 siRNA (Fig. S3b), they did not include the levels of total MKLP1. Indeed, IFT88 siRNA causes a similar reduction in the levels of total MKLP1 ((Fig S1e) and therefore their experiment does not support the interpretation that IFT88 is necessary for Aurora B-mediated phosphorylation of MKLP1. They should either repeat the experiment properly (there are many commercial anti-MKLP1 antibodies generated in various species) or eliminate this statement from their manuscript.

According to the reviewer's suggestion, we removed from the text the sentence: "In agreement with this statement, MKLP1 phosphorylation on S708, which is known to depend on Aurora B, was reduced at the central spindle upon IFT88 depletion (Supplementary Fig. 3b)¹⁵."

The text now says (Page 6): "This defect in Aurora B relocalization likely explains the decrease in MKLP1, phospho S708 MKLP1 (Supplementary Fig. 3b)¹⁵ and MgcRacGAP observed at the central spindle".

The title of supplementary figure 3 has also been modified to remove this statement. The new figure title is: "Aurora B relocalization is impaired and Phospho S708-MKLP1 is decreased at the central spindle upon IFT88 depletion."